# There are no Champions in Long-Term Time Series Forecasting

## Abstract

Recent advances in long-term time series forecasting have introduced numerous complex prediction models that consistently outperform previously published architectures. However, this rapid progression raises concerns regarding inconsistent benchmarking and reporting practices, which may undermine the reliability of these comparisons. Our position emphasizes the need to shift focus away from pursuing ever-more complex models and towards enhancing benchmarking practices through rigorous and standardized evaluation methods. To support our claim, we first perform a broad, thorough, and reproducible evaluation of the top-performing models on the most popular benchmark by evaluating five models over 14 datasets encompassing 3,500+ trained networks for the hyperparameter (HP) searches. Then, through a comprehensive analysis, we find that slight changes to experimental setups or current evaluation metrics drastically shift the common belief that newly published results are advancing the state of the art. Our findings suggest the need for rigorous and standardized evaluation methods that enable more substantiated claims, including reproducible HP setups and statistical testing.

## 1 Introduction

Long-term time series forecasting (LTSF) is critical across various domains, including energy management [58], financial planning [45], and environmental modeling [49]. Accurately predicting future values in time series data enables better decision-making and resource allocation. LTSF remains challenging due to the complex temporal dynamics, including trends, seasonality, irregular fluctuations, and significant variability across datasets [41, 46].

Recent advances in deep learning have improved LTSF capabilities, and the field is currently witnessing an exponential surge in publication rates [28]. Transformer models have been adapted to time series forecasting with innovative modifications, such as univariate patching [39] and attention mechanisms tailored to exploit inter-variate dependencies [34, 57]. In addition, models leveraging multiscale signal mixing [55], and Fourier-based 2D decomposition [60] have expanded the field.

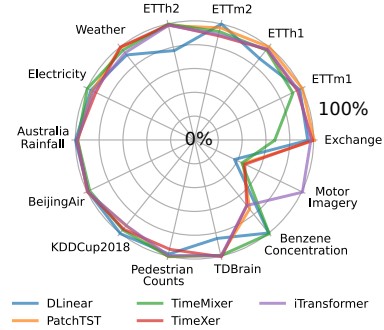

Figure 1: **All models are essentially equivalent.** Results overview in terms of relative MSE averaged over all forecast horizons for the leaderboard models of the popular TSLib benchmark [56].

However, we claim that the field is facing significant challenges regarding fair benchmarking, transparent reporting, and guidelines for model selection. Although this problem is not uniquely encountered in LTSF [21, 15], we highlight the specific challenges in LTSF to promote discussions

towards improvements in the field, as it was done in other disciplines [4, 38, 44, 62]. In LTSF, we observe inconsistencies in test setups across different benchmarks, biased comparisons, and challenges with reproducibility that hinder fair performance assessment in the field. Moreover, marginal performance gains in recent literature cast doubt on the practical value of increasingly complex model architectures [66]. **Our position emphasizes the need to shift focus away from pursuing ever-more complex models and towards enhancing benchmarking practices through rigorous and standardized evaluation methods.** To support our claim, we conduct a comprehensive, rigorous, and reproducible evaluation of the top-performing models on the most widely used benchmark, encompassing five models over 14 datasets (3,500+ trained networks for the HP searches). Our findings reveal that no single model consistently outperforms all the baselines (Fig. 1). This result directly challenges the prevailing narrative of new architectures consistently surpassing competing models across all domains [34, 39, 55, 57, 60]. We analyze the potential reasons behind this phenomenon and propose recommendations to help the field progress. To facilitate ongoing research in LTSF, we will publish our code upon acceptance. The contributions of our paper are as follows:

- We question the narrative of consistently dominated LTSF benchmarks (Sec. 2) supported by results obtained over a comprehensive and reproducible experimental setup (Sec. 3).
- We challenge previous guidelines for model selection based on dataset characteristics (Sec. 3) and highlight the need for further research in this direction (Sec. 5).
- We investigate potential causes behind overstated claims by carefully analyzing our experimental setup and those from prior work in the literature (Sec. 4), and offer recommendations to help prevent similar issues in future work (Sec. 5).

## 2 Field overview

We provide an overview of recent advancements in LTSF, focusing on current benchmarks (Sec. 2.1) and emerging champions (Sec. 2.2). Due to space limitations, additional related work on recent time series forecasting models and paradigms is included in Appendix A.

### 2.1 Benchmarks and their recommendations for dataset-guided model selection

**TSLib** [56] compares 12 deep learning models across five tasks: classification, imputation, anomaly detection, and long-/short-term forecasting. For long-term forecasting, nine datasets from four domains are used. Results are presented for two settings: unified hyperparameters (HPs) and an HP search per model, but details on parameters, context length, forecast horizon, or the search process are missing. The evaluation metric is the mean squared error (MSE) averaged across datasets. The authors claim that their results clearly show the superior forecasting capabilities of transformer models, particularly iTransformer and PatchTST, despite arguably marginal improvements over MLP-based models like N-Beats [40]. They stress the importance of continued exploration of methods that use temporal tokens in time series analysis.

**TFB** [41] evaluates 22 statistical, classical machine learning, and deep learning methods using 25 multivariate and 8,068 univariate datasets. The experiments adhere to each method's HPs specified in the original works. Based on their results, the authors claim that linear models outperform deep learning methods in datasets with increased trends and distribution shifts. Conversely, transformers excel in datasets with marked patterns (e.g., seasonality). Notably, PatchTST and DLinear [66] consistently perform well across datasets, exhibiting no major weaknesses.

**BasicTS+** [46] incorporates 28 forecasting models, including 17 short-term forecasting (STF) and 11 LTSF models, across 14 widely used datasets. STF models encompass prior-graph-based, latent-graph-based, and non-graph-based methods, while LTSF models consist of transformer-based and linear-layer-based architectures. Models are implemented following publicly available architectures and HPs, with further tuning of parameters like learning rate and batch size via grid search to ensure performance is at least as good as reported in the original paper. Upon analyzing the results, the authors argue that dataset characteristics play a major role in determining model performance. They claim that transformer models excel on datasets with clear, stable patterns, whereas simpler models like DLinear perform comparably on datasets without such patterns. The authors emphasize the need to address data distribution drift and unclear patterns instead of focusing solely on increasing model complexity. They suggest that this may indicate potential overfitting to commonly used datasets like

ETT*, *Electricity*, *Weather*, and *Exchange*, which risks creating a misleading impression of progress. They conclude that careful dataset selection and curation are essential to advance the field.

## 2.2 Emergent LTSF champions

Recent models have made a leap in LTSF performance [34, 39, 55, 57, 60, 66]. A popular benchmark for LTSF is TSLib [56], which, at the time of writing, has accumulated over 8k stars and 1.3k forks on GitHub, reflecting its wide adoption in the field. There has been a series of new models in 2024 that reportedly dominate the field. The current leaderboard in TSlib includes five models, originally published in top-tier machine learning conferences. We summarize the striking win percentages from their original works in Tab. 1 and briefly describe each model below.

Table 1: **Model win rates in previous works.** Winners in LTSF for forecast horizons $T \in \mathcal{T} = \{96, 192, 336, 720\}$. The win rates (%) are according to each $T$ without averaging. TimeMixer reported unified parameters (A) and HP search (B) results. † avg. over $\mathcal{T}$ and ETT* datasets.

| Model | Win % (MSE) | Win % (MAE) |
|---|---|---|
| DLinear [66] | 50.0 | 16.7 |
| PatchTST [39] | 87.5 | 59.4 |
| TimeMixer [55] | A) 93.8
B) 81.2 | A) 100
B) 81.2 |
| iTransformer [34] | 33.3
71.4† | 47.2
85.7† |
| TimeXer [57] | 85.7 | 60.7 |

> **TSLib Long-Term Time Series Forecasting Leaderboard**
>
> We present top TSLib models under fixed and searched look-back settings, ordered by publication year:
> [66] **DLinear** - *AAAI-23*: Linear model introduced to challenge transformers in early benchmarks.
> [39] **PatchTST** - *ICLR-23*: Transformer with univariate patching of time series for tokenization.
> [55] **TimeMixer** - *ICLR-24*: MLP-mixer that introduced past-decomposable and future-predictor mixing.
> [34] **iTransformer** - *ICLR-24*: Transformer variant that attends across variates instead of patches.
> [57] **TimeXer** - *NeurIPS-24*: Transformer with dual attention for interactions on patches and variates.

# 3 Who is the real champion?

**Motivating *confirmatory research* in LTSF.** As seen in the previous section, recent papers often suggest that newly proposed architectures outperform others across almost all tested datasets. However, the variability in results for the same algorithms and reliance on prior studies with different experimental setups raise questions about their consistent superiority. This concern goes beyond the field of LTSF and is well supported by a growing body of work criticizing the reliability and generalizability of empirical results in machine learning [4, 15, 26, 44, 48, 50, 62]. In response, the community has increasingly recognized the need for more *confirmatory research* [21], i.e., empirical evaluations conducted by researchers without vested interest in a particular method, aiming to ensure fairness, minimize bias, and reduce overly optimistic conclusions. Motivated by the exponential surge in publication rates in time series forecasting [28] and the community quest for more neutral benchmarking [21], we selected the previously introduced five top-performing models from TSLib [56] and performed a comprehensive empirical evaluation across 14 datasets from various domains. For completeness, a comparison including classical statistical methods is presented in Appendix H.

**Datasets.** We evaluate models on 14 datasets from five domains: Energy, Economy, Transport, Health, and Environment. They vary substantially in sample size, sampling frequency, number of variates, and various statistics for time series dynamics (Appendix B). The selection aims to minimize bias by preventing any model from gaining an unfair advantage due to specific dataset characteristics.

**HP search.** We searched HPs aligned with those described in TimeMixer [55]. Specifically, we optimized input lengths, learning rates, and the number of encoder layers. We used the Optuna framework [1], employing the default `TPEsampler` for HP sampling and the `SuccessiveHalvingPruner` as the trial scheduler. We employed Adam with an exponentially decaying scheduler. The optimized HPs were used to train and evaluate the final model across three random seeds to ensure robust results. For the rest of the model HPs, we default to the configurations provided in TSLib. We refer the reader to Appendix C for a more detailed description.

Table 2: **Results.** Mean and best (i.e., Min) values averaged over prediction lengths. Datasets span the energy, economy, transport, health, and environment domains. **Best** and second-best are highlighted.

| Model | DLinear | | | | PatchTST | | | | iTransformer | | | | TimeMixer | | | | TimeXer | | | |
|---|---|---|---|---|---|---|---|---|---|---|---|---|---|---|---|---|---|---|---|---|
| Metric | MAE | | MSE | | MAE | | MSE | | MAE | | MSE | | MAE | | MSE | | MAE | | MSE | |
| Statistic | Mean | Min | Mean | Min | Mean | Min | Mean | Min | Mean | Min | Mean | Min | Mean | Min | Mean | Min | Mean | Min | Mean | Min |
| ETTh1 | 0.477 | 0.476 | 0.474 | 0.472 | 0.432 | 0.428 | 0.414 | 0.408 | 0.443 | 0.441 | 0.424 | 0.422 | 0.443 | 0.439 | 0.429 | 0.424 | 0.44 | 0.436 | 0.425 | 0.42 |
| ETTm1 | 0.422 | 0.421 | 0.403 | 0.402 | 0.431 | 0.429 | 0.394 | 0.391 | 0.437 | 0.433 | 0.408 | 0.403 | 0.449 | 0.439 | 0.432 | 0.416 | 0.445 | 0.44 | 0.412 | 0.405 |
| ETTh2 | 0.474 | 0.474 | 0.485 | 0.484 | 0.41 | 0.406 | 0.379 | 0.374 | 0.405 | 0.403 | 0.378 | 0.374 | 0.404 | 0.401 | 0.374 | 0.368 | 0.406 | 0.404 | 0.377 | 0.375 |
| ETTm2 | 0.261 | 0.261 | 0.151 | 0.151 | 0.269 | 0.268 | 0.156 | 0.155 | 0.276 | 0.275 | 0.166 | 0.165 | 0.27 | 0.269 | 0.162 | 0.16 | 0.277 | 0.274 | 0.168 | 0.165 |
| Electricity | 0.259 | 0.258 | 0.162 | 0.161 | 0.261 | 0.261 | 0.164 | 0.163 | 0.262 | 0.258 | 0.165 | 0.162 | 0.254 | 0.252 | 0.156 | 0.154 | 0.268 | 0.263 | 0.17 | 0.165 |
| Weather | 0.298 | 0.298 | 0.244 | 0.244 | 0.263 | 0.262 | 0.225 | 0.223 | 0.279 | 0.278 | 0.238 | 0.237 | 0.271 | 0.264 | 0.231 | 0.225 | 0.264 | 0.262 | 0.224 | 0.222 |
| Exchange | 0.434 | 0.37 | 0.389 | 0.265 | 0.409 | 0.405 | 0.368 | 0.362 | 0.416 | 0.409 | 0.379 | 0.364 | 0.476 | 0.409 | 0.55 | 0.38 | 0.412 | 0.409 | 0.374 | 0.37 |
| MotorImagery | 1.183 | 1.182 | 4.592 | 4.583 | 1.006 | 1 | 3.775 | 3.745 | 0.383 | 0.338 | 1.692 | 1.496 | 1.011 | 0.996 | 3.869 | 3.813 | 0.915 | 0.893 | 3.649 | 3.576 |
| TDBrain | 0.802 | 0.801 | 1.151 | 1.15 | 0.725 | 0.722 | 0.982 | 0.976 | 0.722 | 0.721 | 0.978 | 0.974 | 0.723 | 0.72 | 0.981 | 0.974 | 0.719 | 0.717 | 0.97 | 0.965 |
| BeijingAir | 0.472 | 0.471 | 0.583 | 0.582 | 0.46 | 0.458 | 0.578 | 0.576 | 0.464 | 0.461 | 0.58 | 0.572 | 0.471 | 0.467 | 0.592 | 0.58 | 0.462 | 0.459 | 0.582 | 0.577 |
| BenzeneConcentration | 0.02 | 0.018 | 0.008 | 0.008 | 0.042 | 0.038 | 0.011 | 0.011 | 0.053 | 0.048 | 0.012 | 0.011 | 0.022 | 0.019 | 0.008 | 0.008 | 0.037 | 0.032 | 0.012 | 0.011 |
| AustraliaRainfall | 0.751 | 0.751 | 0.838 | 0.838 | 0.757 | 0.756 | 0.855 | 0.852 | 0.754 | 0.753 | 0.849 | 0.848 | 0.755 | 0.754 | 0.853 | 0.85 | 0.753 | 0.752 | 0.847 | 0.846 |
| KDDCup2018 | 0.63 | 0.627 | 0.997 | 0.989 | 0.647 | 0.644 | 1.086 | 1.076 | 0.648 | 0.646 | 1.088 | 1.084 | 0.631 | 0.627 | 1.035 | 1.03 | 0.64 | 0.633 | 1.045 | 1.03 |
| PedestrianCounts | 0.289 | 0.289 | 0.298 | 0.298 | 0.293 | 0.291 | 0.291 | 0.289 | 0.285 | 0.283 | 0.295 | 0.291 | 0.284 | 0.279 | 0.292 | 0.286 | 0.307 | 0.295 | 0.311 | 0.295 |
| Average | 0.484 | 0.478 | 0.77 | 0.759 | 0.458 | 0.455 | 0.691 | 0.686 | 0.416 | 0.41 | 0.547 | 0.529 | 0.462 | 0.452 | 0.712 | 0.691 | 0.453 | 0.448 | 0.683 | 0.673 |
| Rank | 3.14 | 3.07 | 3.29 | 3.21 | 2.79 | 3.07 | 2.5 | 2.64 | 3.14 | 3.29 | 3.14 | 3.07 | 3 | 2.64 | 3.07 | 2.93 | 2.93 | 2.93 | 3 | 3.14 |

**Finding: There is no champion.** We follow the TSlib benchmark and use MSE and mean absolute error (MAE) to evaluate model performance. Tab. 2 presents the MSE and MAE for each dataset, averaged over the most common forecast horizons [34, 39, 57], revealing results that differ substantially from recent papers where proposed algorithms often dominate. Instead, our findings indicate no definitive best-performing model across all datasets and forecast horizons (Tab. 12 and Tab. 13, Appendix H). To assess reliability, we compared our results with the best-reported outcomes from the original studies on three common datasets—*ETT\**, *Electricity*, and *Weather*—and found that our HP search performed similarly or better, confirming the proper implementation and tuning of our baselines (Tab. 9, Appendix D). To present a comprehensive view that highlights both the optimal outcomes and the realistic performance variability, we report the minimum values (best MSE/MAE) and the averages. We further analyze HP-search run-to-run variability in Appendix E.

**Finding: Recommendations for dataset-guided model selection do not hold.** We analyze whether our results align with the guidelines proposed by BasicTS+ [46] and TFB [41] (Sec. 2.1), which aim to address the challenge of selecting appropriate models for specific datasets in LTSF.

We revisit the example in [46] and assess the performance of a linear model (DLinear) versus a transformer model (PatchTST) on data with clear and unclear patterns, respectively. We use *Exchange* as the dataset with an unclear pattern and *Pedestrian-Counts* as the dataset with a clear pattern (Fig. 5, Appendix B.2). PatchTST outperforms DLinear on both occasions (Tab. 12, Appendix H), contradicting the previous recommendations that linear models should be preferred when the data lacks clear patterns [46]. Next, to assess guidelines regarding uni-

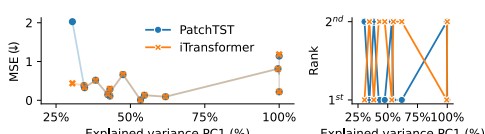

Figure 2: **Univariate vs. multivariate** PatchTST and iTransformer perform comparably as a function of explained variance.

variate and multivariate data, we compare PatchTST (univariate) against iTransformer (multivariate) on all datasets. We use the explained variance from principal component analysis as a proxy for inter-variate similarity (Appendix B.1). Neither model performs increasingly better depending on the inter-variate similarity as ranks fluctuate across the spectrum (Fig. 2), contradicting the guideline that multivariate models should be preferred if the data has strong inter-variate similarities [41, 46].

## 4 Why are they all champions?

### 4.1 Impact of selective inclusion of datasets and forecast horizons

**Model rankings are highly sensitive to dataset and horizon selection.** We systematically analyze how the selective exclusion of datasets and forecast horizons in the experimental settings may affect overall rankings. Let $\mathcal{D} = \{D_1, \ldots, D_M\}$ be the collection of datasets and $\mathcal{T} = \{T_1, \ldots, T_H\}$ the forecast horizons. We sample $K = 5,000$ experimental configurations, each defined by uniformly drawn subsets of datasets and horizons: $\mathcal{S}_D \subseteq \mathcal{D}$, $|\mathcal{S}_D| = k$, $k \sim \mathcal{U}\{1, M\}$, and $\mathcal{S}_T \subseteq \mathcal{T}$, $|\mathcal{S}_T| =$

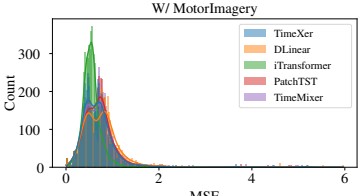
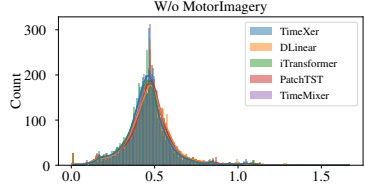

| W/ *MotorImagery* | Win % | MSE |
|---|---|---|
| iTransformer | 59.86 | **0.55 ± 0.23** |
| PatchTST | 13.34 | 0.72 ± 0.44 |
| TimeXer | 10.20 | 0.70 ± 0.44 |
| DLinear | 9.20 | 0.78 ± 0.53 |
| TimeMixer | 7.40 | 0.72 ± 0.45 |
| **W/o *MotorImagery*** | **Win %** | **MSE** |
| iTransformer | 15.80 | **0.46 ± 0.14** |
| PatchTST | 30.64 | 0.47 ± 0.15 |
| TimeXer | 21.08 | **0.46 ± 0.13** |
| DLinear | 14.96 | 0.48 ± 0.15 |
| TimeMixer | 17.52 | 0.47 ± 0.14 |

Figure 3: **Model rankings are highly sensitive to dataset and horizon selection.** We assess the robustness of rankings across 5,000 experimental configurations, each using a random subset of datasets and forecast horizons. Including *MotorImagery*, the only dataset with clear model gaps (Fig. 7, Appendix H), favors iTransformer, while excluding it yields close performance across models. This highlights the brittleness of current benchmarks, where small changes in datasets or forecast horizons can easily shift which model appears as a champion. **Best** and second-best are highlighted.

$\ell$, $\ell \sim \mathcal{U}\{1, H\}$. When all datasets are included, iTransformer seems to clearly be the best model, obtaining a win percentage of ∼60% and the lowest average MSE of 0.55 over the distribution (Fig. 3, left). However, after repeating the analysis without *MotorImagery*, i.e., the only dataset with a clear performance gap among baselines (Fig. 7, Appendix H), the distributions overlap (Fig. 3, center), implying equivalency of all models. Since win percentages and average MSEs are similar across models, minor changes in datasets and forecast horizons can shift which model appears best, supporting our claim regarding the brittleness of current benchmarks and model superiority claims.

**Prior work employed inconsistent dataset and horizon selection.** We observe that subsets of the full benchmark were occasionally used, which may be justified, e.g., for lack of dataset size. In iTransformer [34], the authors averaged the performance over the ETT datasets, which they justified by their intrinsic similarity. However, this increased the percentage of wins of their proposed method from 33.3% to 71.4%. BasicTS+ [46] focuses solely on a forecast horizon of 336, whereas TFB [41] bases the analysis of the impact of different data characteristics on a horizon of 96, despite reporting performance for all four forecast horizons.

## 4.2 Impact of selective inclusion of baseline models

**Model exclusion reshape leaderboards.** While unsurprising, we stress that excluding the top model from a benchmark may automatically crown the second-best as a champion. For instance, removing iTransformer from Tab. 2 would champion TimeXer as it scores the second-best average metrics.

**Prior work overlooked SOTA models in the analyses.** We notice the lack of inclusion of baselines like N-Beats in the publications of recent "champions", although it is a top-3 method in [56] (Appendix G). In addition, we identified cases where the best-performing model was excluded from discussions without justification. For example, [41, 46] claim recent transformers underperform compared to earlier methods, but their experiments show the most recent LTSF transformer at the time (PatchTST) outperformed competitors, contradicting this claim. Moreover, [46] claims linear models are better for LTSF on datasets with unclear patterns or distribution shifts. However, the claim is based on experiments that excluded PatchTST, thereby weakening the strength of their conclusion.

## 4.3 Impact of hyperparameters

**HP search raises absolute performance.** The impact of HP tuning on the benchmarking performance of models is becoming increasingly evident [7, 37]. We investigate whether this is also the case in LTSF via a proof-of-concept example. We report the evaluation in terms of MSE for DLinear on the *Weather* dataset at forecast horizon 96 (Tab. 3). HP tuning brings a relative performance boost of ∼15% in our setup and an ∼10% in [56]. Similarly, building on the HP search details in [57], we found comparable performance between TimeMixer, iTransformer, and PatchTST, unlike the original work, where TimeMixer consistently ranks first. This underscores how close the actual performance of models is, making outcomes and conclusions sensitive to slight variations in HP search.

**Prior work put models at disadvantage through unified HP setup.** In TSLib, models are usually based on the implementation of the original publications. However, in [60], for a fair comparison, they changed the input embeddings and the final projections to be the same for all models. Specifically,

Table 3: **HP search sensitivity.** We report the MSE of DLinear for *Weather* at prediction length 96 when HP tuning is/is not performed, both in our and previous papers, along with the **relative performance improvement** expressed in % (when possible). "−" indicates a missing analysis.

| DLinear (MSE) | [66] | [39] | [60] | [34] | [55] | [57] | Ours |
|---|---|---|---|---|---|---|---|
| Unified HP | – | – | 0.196 | 0.196 | 0.195 | 0.196 | 0.198 |
| HP tuning | 0.176 | 0.176 | – | – | 0.176 | – | 0.168 |
| Rel. Improv. | – | – | – | – | **+9.7%** | – | **+15.1%** |

the sequence length was set to 96 for all models by default. This is critical since DLinear [66], after a broad ablation study, explicitly states that short input sequences ($< 336$) lead to underfitting.

### 4.4 Influence of visualizations

**Visualizations may bias perceived rankings.** Visualizations are a strong tool to convey a message. In Fig. 4, we investigate the impact of scales to visualize results. We observe that using an absolute scale in radar plots exaggerates differences between models that are not perceived in the relative scale with uniform axes. Conversely, it can obscure substantial differences, as in the example of *MotorImagery* and *BenzeneConcentration* (Fig. 4, right, b and e).

**Prior work employed misleading visualization practices.** We identified two examples that used radar plots with absolute scales to visualize the model performances between models [34] and across dataset characteristics [41], respectively. These choices can create a misleading impression of the models' actual performance and lead to false conclusions. Note that other plots may also create biased impressions through axis scaling or selective metric representation.

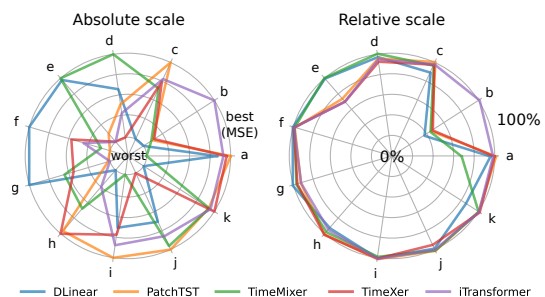

a) Exchange, b) MotorImagery, c) ETT, d) Electricity, e) BenzeneConcentration, f) AustraliaRainfall, g) KDDCup2018, h) Weather, i) BeijingAir, j) PedestrianCounts, k) TDBrain

Figure 4: **Bias in visualizations.** The plots show the same results (MSE) represented at different scales. The relative scale makes performance differences between models appear more subtle.

### 4.5 Statistical evidence for model superiority

**Analysis.** We illustrate, through a proof-of-concept example, how the lack of robust statistical testing can lead to false claims regarding models' superiority. First, we introduce recommended non-parametric statistical tests [13]. Second, we upgrade the existing iTransformer to emulate current model-design proposals and rigorously evaluate it in our setup (see Sec. 3 for details).

**Statistical tests.** [13] studied various statistical tests for comparing classifiers from both theoretical and empirical perspectives. The study recommended a set of simple, reliable, and robust tests for such comparisons. In particular, the sign test [43] compares two classifiers over multiple datasets, and the Friedman test compares various classifiers over multiple datasets (see Appendix F for details).

**iPatch: A proof-of-concept model.** We briefly introduce the iPatch model as a hybrid architecture that integrates design principles of PatchTST into iTransformer, emulating common model design trends observed in recent research. Let the input series be $\mathbf{x} \in \mathbb{R}^{B \times C \times L}$, where $B$ is the batch size, $C$ is the number of variates, and $L$ is the lookback length. Firstly, in iPatch, unlike iTransformer, we reshape the input by splitting the sequence into $N$ cycles of length $P$ ($L = N \cdot P$), transforming it from $B \times C \times L$ to $B \times (C \cdot N) \times P$ to later enable in the layer the temporal attention as in PatchTST. Each cycle is subsequently embedded to $d_m$, resulting in $\mathbf{z} \in \mathbb{R}^{B \times (C \cdot N) \times d_m}$. Secondly, we enhance the attention module as a sequence of two attentions over variates and cycles. First, the input is reshaped to $(B \cdot N) \times C \times d_m$ to isolate the $C$ variates for each cycle and consequently to

Table 4: **iPatch as a questionable champion.** Although iPatch scores the **best** average rank and the second-best MSE/MAE averaged over all datasets, it does not statistically differ from all the other baselines under a Friedman test.

| Model | MSE | MAE | Rank (MSE) |
|---|---|---|---|
| DLinear | 0.770 | 0.484 | 3.79 |
| PatchTST | 0.691 | 0.458 | **3.07** |
| iTransformer | **0.547** | **0.416** | 3.93 |
| TimeMixer | 0.712 | 0.462 | 3.64 |
| TimeXer | 0.683 | 0.453 | 3.50 |
| iPatch | 0.580 | 0.422 | **3.07** |

apply attention over $C$ similarly to iTransformer. Next, the output of the variate attention is reshaped to $(B \cdot C) \times N \times d_m$ to isolate temporal cycles and apply the second attention mechanism similarly to PatchTST. Since the iTransformer MLPs operating on univariate data were hypothesized to capture time series properties like amplitude and periodicity [34], we reshape the series to $(B \cdot N) \times C \times d_m$ before applying the MLP and finally map it back to $B \times (C \cdot N) \times d_m$ for the next transformer layer.

**Statistical testing substantiates performance gains from targeted architectural adjustments.** Initially, we evaluate the performance of iPatch following either average MSE/MAE [56] or ranks. Tab. 4 shows that iPatch achieves the best average rank and the second-best MSE and MAE. Complete results for iPatch are available in Tab. 14, Appendix H. In line with these outcomes and common practices in the field of LTSF, iPatch may "almost" seem the best-performing model. However, performing the Friedman test, we observe that no model excels over the others ($\chi^2_F = 1.14$, $p = 0.95$). Given the lack of a clear winner, we conduct a focused comparison between iPatch and iTransformer using the sign test [43], since iPatch builds directly upon iTransformer. Despite their similar architectures, iPatch wins in terms of MSE/MAE on 11 out of 14 datasets, yielding a statistically significant p-value of 0.05 under the sign test (Appendix F), suggesting that targeted architectural modifications can lead to performance improvements.

**Prior work neglected statistical testing.** The TSLib benchmark [56] employs averaging for presenting aggregated results. However, averages are susceptible to outliers [13]. A classifier's strong performance on one dataset can mask weaknesses elsewhere, so we prioritize consistent performance across problems, making dataset averaging unsuitable for evaluation (Sec. 4). BasicTS+ [46] and TFB [41] focus on the number of wins achieved by each model but do not incorporate any statistical testing, making conclusions less reliable and harder to communicate in a concise manner.

## 4.6 Trade-off between model performance and efficiency

**Analysis.** It is also essential to consider other factors that contribute to a model's superiority, such as the trade-offs between performance (e.g., MSE) and efficiency (e.g., training speed or memory consumption) introduced by architectural modifications.

Table 5: **Limited gains from increased model complexity.** Efficiency-weighted performance comparison ($\xi$) relative to DLinear ($\uparrow$ is better). Despite higher complexity, most models fail to outperform DLinear across multiple efficiency metrics with all datasets (A) and without *MotorImagery* (B). TP indicates throughput. **Best** and second-best are highlighted.

| $\xi(m, \texttt{DLinear}, \texttt{MSE}, \Phi)$ ($\uparrow$) | DLinear | | PatchTST | | iTransformer | | TimeMixer | | TimeXer | |
|---|---|---|---|---|---|---|---|---|---|---|
| | A | B | A | B | A | B | A | B | A | B |
| $\Phi = \{\texttt{FLOPs}\}$ | 1.00 | **1.00** | 0.86 | 0.79 | **1.21** | 0.90 | 0.69 | 0.65 | 0.78 | 0.71 |
| $\Phi = \{\texttt{\#params}\}$ | 1.00 | **1.00** | 0.95 | 0.88 | **1.27** | 0.94 | 0.95 | 0.89 | 0.95 | 0.87 |
| $\Phi = \{\texttt{TP}, \texttt{memory}\}$ (train) | 1.00 | **1.00** | 0.93 | 0.86 | **1.32** | 0.97 | 0.81 | 0.77 | 0.92 | 0.84 |
| $\Phi = \{\texttt{TP}, \texttt{memory}\}$ (test) | 1.00 | **1.00** | 0.95 | 0.87 | **1.26** | 0.93 | 0.86 | 0.81 | 0.94 | 0.87 |

**Efficiency-weighted error metric.** Drawing inspiration from neural-architecture-search literature [53], we define a composite metric that summarizes prediction quality and efficiency relative to a baseline model. Let $m$ denote a candidate model and $b$ our baseline, with $\epsilon(\cdot)$ representing our prediction error metric and $\Phi = \{\phi_k(\cdot)\}_{k=1}^K$ being a set of efficiency metrics. The efficiency-weighted error metric $\xi$ is formulated as: $\xi(m, b, \epsilon, \Phi) = \frac{\epsilon(b)}{\epsilon(m)} \cdot \prod_{k=1}^K \frac{\phi_k(b)}{\phi_k(m)}^{s_k w}$ where $s_k \in \{-1, +1\}$ controls the metric-specific ratio directionality (lower/higher is better) and $w$ encodes the relative importance of efficiency. Thus, the weighted product formulation of $\xi$ ensures models' error ratios against the baseline are scaled by efficiency ratios weighted by the exponent $w$. When $w = 1$, the efficiency ratio scales linearly with the error ratio, while if $w = 0$, the efficiency ratio has zero relevance. Models with $\xi > 1$ outperform the baseline, while $\xi < 1$ indicate unfavorable trade-offs. The baseline always scores a value of one, i.e., when $m = b$, $\xi = 1$. Furthermore, $b$ is chosen such that $0 < \frac{\phi_k(b)}{\phi_k(m)}^{s_k} < 1$, ensuring that the efficiency term always penalizes all other models unless compensated by accuracy gains.

**Performance-efficiency rankings show lack of improvements.** For this analysis, we consider DLinear as the baseline, being the most efficient model across all metrics. Furthermore, we set $w = 0.07$ so that a 10% reduction in any efficiency metric corresponds approximately to a 1% loss in the error-metric ratio $\frac{\epsilon(b)}{\epsilon(m)}$ when the efficiency ratio $\frac{\phi_k(b)}{\phi_k(m)}$ lies in the range $[0.3, 1]$, as $x^{0.07} \approx 0.1x + 0.9$ in this interval. We select FLOPs and #params for theoretical complexity, and throughput (TP) with memory usage for practical hardware efficiency. In Tab. 5, we report that, with all datasets included (column A), all models except iTransformer underperform DLinear ($\xi < 1$). When the *MotorImagery* dataset is excluded (see Sec. 4.1), even iTransformer no longer achieves superior performance. These results suggest that the additional architectural complexity does not

currently lead to meaningful benefits. This pattern is clearly captured by efficiency-weighted error metrics such as $\xi$, which summarize trade-offs between accuracy and efficiency across all datasets and horizons. Further analyses and details on efficiency metric estimation are provided in Appendix I.

**Prior work lacks consensus on performance-efficiency trade-offs.** Although there are examples that perform certain trade-off analyses [34, 55, 46, 57], their interpretations appear to be limited by selected subsets of datasets, models, and efficiency metrics, lacking a holistic perspective offered by metrics aggregated across all datasets and horizons, thereby conveying incomplete conclusions.

## 5 How can the field establish real champions?

Concluding the previous chapters, we provide recommendations that can be tackled by the community. To ensure continued progress in LTSF, the field must address persistent shortcomings in benchmarking, evaluation methodology, and guidelines for model selection. We outline guidelines aimed at improving transparency, rigor, and the practical relevance of LTSF research.

**Improving benchmarking practices.** To prevent evaluations from disproportionately favoring specific model architectures based on dataset characteristics, benchmark maintainers should curate datasets that are diverse and representative of real-world variability across domains. In that regard, a crucial step is to define meaningful forecast horizons based on intrinsic dataset characteristics—an issue exemplified by the arbitrary performance on datasets such as *Exchange* [22]. Furthermore, benchmarks should provide rigorously tuned HP configurations for all models, ideally supported by integrated HP optimization tools. Benchmark users must report performance consistently across all supported datasets, forecast horizons, and context lengths. Even minor deviations in experimental setup can dramatically shift performance rankings (Sec. 4), underscoring the need for transparency and standardization. Additionally, the field would benefit from objective, independent evaluations, in which test sets are withheld and assessed by third parties, e.g., as originally practiced for ImageNet.

**Reducing unsubstantiated claims.** Researchers should adopt robust statistical testing to supplement performance rankings and mitigate unreliable claims, as exemplified in Sec. 4.5. Visualizations must be designed with care to avoid distorting perceived differences. For instance, scale choices can easily exaggerate or obscure performance gaps as highlighted in Sec. 4.4.

**Revising guidelines for model selection.** To develop effective model selection guidelines, future studies should focus on datasets where performance varies significantly among SOTA models. As illustrated in Fig. 1, only two datasets—*BenzeneConcentration* and *MotorImagery*—exhibit clearly distinguishable performance patterns, highlighting the need for further investigation into what dataset characteristics drive such differences. To advance this goal, the field should conduct comprehensive evaluations across datasets with diverse characteristics and consistently compare a broad range of model architectures, ensuring that the best-performing architecture for each category, such as linear, MLP, and transformer models, is clearly reported. Additionally, practical trade-offs between model performance and efficiency should be assessed by systematically analyzing how architectural changes impacts computational cost and memory usage (Sec. 4.6). Composite metrics such as $\xi$ can unify performance and efficiency into a single score. To support such evaluations, benchmarks should provide standardized functionalities for consistent and detailed comparisons.

| Summary of Recommendations | |
|---|---|
| *Improving benchmarking practices* | · Benchmarks should include datasets that reflect real-world diversity.
· Benchmarks should define forecast horizons informed by dataset characteristics.
· Results should be reported across all datasets and forecast horizons (Sec. 4.1).
· Results from the best-performing models should always be included (Sec. 4.2).
· Rigorously tuned HP configurations must be used for all models (Sec. 4.3).
· Third-party evaluations should be encouraged to strengthen reliability. |
| *Reducing unsubstantiated claims* | · Visualizations should not exaggerate minor differences (Sec. 4.4).
· Statistical tests should be used when comparing models (Sec. 4.5). |
| *Revising guidelines for model selection* | · Methodologies relating model-dataset characteristics should be further explored.
· Performance-efficiency trade-offs should be tackled systematically (Sec. 4.6). |

# 6   Alternative views

**Experimental design has limitations and lacks completeness.** Critics might argue that our results are suboptimal due to experimental limitations, that focusing on recent models excludes influential earlier architectures and introduces bias, or that testing selected models overlooks the field's diversity. However, such criticisms reinforce our core argument: small variations in experimental setups can lead to markedly different model rankings, highlighting the need for standardized and transparent evaluation practices. Our goal was not to establish exhaustive benchmarks or definitive rankings but to show that recent advancements often provide minimal improvement. By focusing on recent models, we intentionally highlight the current state of the field and its challenges in reliably evaluating and comparing new methods. Moreover, while some models may excel in narrow, context-specific scenarios (e.g., iTransformer in *MotorImagery*), such isolated successes do not translate into universal applicability, further supporting our argument against the "champion" narrative.

**The definition of long-term forecasting is arbitrary.** Alternative views may argue, as we also acknowledge, that using fixed forecast horizons for datasets of different domains can render the notion of "long-term" arbitrary and detached from domain-specific constraints. Although not necessarily optimal, the forecast horizons chosen in our experimental setup reflect current practice and enable comparability with past work. In particular, we adopt the horizons used in TSLib, which are consistent with the ones in other recent benchmarks such as BasicTS+ and TFB, further aligning with the ranges reported in the original publications. Identifying truly meaningful forecast horizons remains an open challenge, complicated by the unclear distinction between short- and long-term forecasting. Our findings may also apply to shorter horizons, though this requires empirical testing.

**Forecasting research is moving towards foundation models.** While recent trends in time series research increasingly explore the development of foundation models [47, 65], we argue that this does not diminish the value of critically examining novel architectures. On the contrary, many architectural components and design choices in foundation models are built directly on insights from earlier foundational research. Our work contributes to this ongoing evolution by providing benchmarking and actionable recommendations that support more informed model development. To acknowledge the growing interest in this direction, we included a brief overview of recent developments in foundation models for time series in Appendix A.4. As seen in computer vision, where novel architectures continue to emerge alongside foundation models [2, 70], we anticipate parallel progress in time series research, with both directions reinforcing each other.

**Multimodal LLMs will make specialized forecasting models obsolete.** As outlined in [25], large language models (LLMs) may serve in three roles (R): as Enhancers (R1), which incorporate domain-specific external knowledge while relying on specialized models for prediction; Forecasters (R2), which replace expert models entirely and cast LLMs directly as predictive models; or Agents (R3), which orchestrate workflows involving external tools and models. Considering current evidence of missing substantial improvements for R2 [52, 63] and that R1 and R3 complement rather than replace classical forecasting models, we believe the research direction investigating architecture design and benchmarking will likely move forward in parallel. To reflect the growing interest in LLM-based approaches specifically, we provided a brief overview of relevant work in Appendix A.3.

# 7   Conclusions

In this work, we critically evaluate LTSF research and put forward a proposal to address persistent issues in the field. Importantly, our aim is not to criticize prior work in this longstanding and recently revitalized domain but to provide a constructive analysis that supports both our own work and future research in the field, including its translation into domain-specific applications. Through an extensive and reproducible evaluation of five models across 14 datasets, we demonstrated that claims of consistent performance improvements in newly published models often rely on specific experimental setups and evaluation methods. Our findings question the idea of universal advancements, revealing that no single model consistently excels across our experiments. We identified issues in the LTSF domain, such as non-standardized evaluation frameworks, biased comparisons, and limited reproducibility that hinder fair assessment and delay real progress. To address these challenges, we propose a set of actionable recommendations: adopt standardized evaluation protocols, prioritize benchmarking robustness over architectural complexity, and deepen the analysis of how dataset characteristics influence model performance.

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

# A  Related work

## A.1  Classical approaches

Traditional statistical methods, such as AutoRegressive Integrated Moving Average [5], Vector Autoregression [54], Exponential Smoothing [24], and Spectral Analysis [29] were widely used in TS forecasting. Progressively, machine learning models such as XGBoost [11], Random Forest [6], Gradient Boosting Regression Trees [16], and LightGBM [27] have shown improvements in the forecast due to their ability to handle non-linear patterns.

## A.2  Deep learning models

Deep learning models have advanced TS forecasting, starting with Recurrent Neural Networks (RNNs), specifically designed to model sequential data. In particular, advanced variants such as RNNs with Long Short-Term Memory units, widely adopted within the TS community, have seen significantly increased usage [23]. Additionally, MLP-based models, such as DLinear [66], N-BEATS [40], and N-Hits [9] use MLP to learn the coefficients that produce both backcast and forecast outputs from their structure.

Originally from Natural Language Processing (NLP), the Transformer architecture is increasingly adapted for time series forecasting, often with modified attention layers to capture temporal dependencies, as seen in Sec. 2 and other prior works, which we describe in the following. Informer [68] and Pyaformer [33] are transformer-based models that modify the attention mechanism. Informer designs a ProbSparse self-attention mechanism to replace the standard self-attention. Pyaformer, on the other hand, presents a pyramidal attention module, where the inter-scale tree structure captures features at different resolutions, and the intra-scale neighboring connections model the temporal dependencies across different ranges. Wu et al. [61] introduced the Autoformer with an Auto-Correlation mechanism to capture the series-wise temporal dependencies based on the learned periods. Following, FEDformer [69] utilizes a mixture-of-expert framework to improve seasonal-trend decomposition and integrates Fourier and Wavelet-enhanced blocks to capture key structures in the TS. [67] presented Crossformer, a transformer-based model utilizing cross-dimension dependency for multivariate TS forecasting. Another recent approach is TimesNet [60], which is a univariate 2D CNN that segments 1D time series according to Fourier decomposition. The segments are then stacked to build a 2D series. This enables the convolutions to simultaneously look at the local structure of the signal at $t_i$ and $t_{i-T}$ simultaneously, where $T$ denotes a dominant signal period.

## A.3  Large Language Models

The success of Large Language Models (LLMs) like BERT and GPT in NLP has inspired researchers to apply these models to TS tasks. One significant approach involves transforming numerical TS data into natural language prompts to leverage pre-trained language models without modifications. PromptCast [64] and [20] present this method, demonstrating generalization in zero-shot settings and often outperforming traditional numerical models. However, recent work cast doubts on the actual significance of LLMs as base forecasters [52]. Moving to few-shot training strategies, TEST [51] adapts TS data for pre-trained LLMs by tokenizing the data and aligning the embedding space, particularly in few-shot and generalization scenarios. Several frameworks focus on enhancing TS forecasting through specialized fine-tuning strategies such as LLM4TS [10] and TEMPO [8].

## A.4  Foundation Models

There is a growing interest in foundation models designed explicitly for TS tasks [47, 65]. Tiny Time Mixers [14] introduce a compact model for multivariate TS forecasting. Timer-XL is a foundation model for unified time series forecasting, supporting univariate and multivariate data by extending next-token prediction for causal generation [36]. The model introduced a universal TimeAttention mechanism to capture fine-grained intra- and inter-series dependencies. MOIRAI [59] addresses challenges like cross-frequency learning and varied distributional properties in large-scale data, achieving competitive zero-shot forecasting performance. TimeGPT-1 [19] and Lag-LLama [42], utilizing decoder-only transformer architectures and achieving strong zero-shot generalization. Chronos [3] trains transformer-based models on discrete tokens processed from TS data, demonstrating superior performance on diverse datasets.

Table 6: **Dataset statistics.** Refer to Appendix B.1 for a detailed description of the statistics.

| Domain | Dataset | # Timesteps | # Channels | Shannon Entr. | Spectral Entr. | Sample Entr. | Stationarity | Complexity | Expl. Var. | Source |
|--------|---------|-------------|------------|---------------|----------------|--------------|--------------|------------|------------|--------|
| Energy | ETTh1 | 17420 | 7 | 0.775 | 0.669 | 0.769 | -5.909 | 0.497 | 0.344 | [56] |
| Energy | ETTm1 | 69680 | 7 | 0.789 | 0.548 | 0.430 | -14.985 | 0.485 | 0.346 | [56] |
| Energy | ETTh2 | 17420 | 7 | 0.813 | 0.639 | 0.526 | -4.136 | 0.397 | 0.431 | [56] |
| Energy | ETTm2 | 69680 | 7 | 0.817 | 0.527 | 0.319 | -5.664 | 0.425 | 0.431 | [56] |
| Energy | Electricity | 26304 | 321 | 0.516 | 0.497 | 0.714 | -8.445 | 0.673 | 0.547 | [56] |
| Environment | Weather | 52696 | 21 | 0.453 | 0.57 | 0.110 | -26.681 | 0.632 | 0.424 | [56] |
| Economic | Exchange | 7588 | 8 | 0.805 | 0.347 | 0.066 | -1.902 | 0.529 | 0.618 | [56] |
| Health | MotorImagery | 1134000 | 64 | 0.719 | 0.519 | 0.326 | -3.133 | 0.763 | 0.305 | [36] |
| Health | TDBrain | 2221212 | 33 | 0.823 | 0.749 | 0.987 | -3.167 | 0.404 | 0.475 | [36] |
| Environment | BeijingAir | 407184 | 9 | 0.493 | 0.686 | 0.951 | -13.253 | 0.165 | 0.383 | [36] |
| Environment | BenzeneConcentration | 2042880 | 8 | 0.799 | 0.701 | 1.938 | -3.114 | -0.049 | 0.534 | [36] |
| Environment | AustraliaRainfall | 3846408 | 3 | 0.838 | 0.604 | 2.215 | -31.734 | -0.013 | 0.996 | [36] |
| Environment | KDDCup2018 | 2942364 | 1 | 0.569 | 0.665 | 0.410 | -9.379 | 0.530 | 1.000 | [36] |
| Transport | PedestrianCounts | 3132346 | 1 | 0.687 | 0.522 | 0.412 | -4.590 | 0.630 | 1.000 | [36] |

## B  Datasets

We include a popular set of datasets (*ETT\**, *Electricity*, *Weather*, *Exchange*) and a set of larger datasets (*MotorImagery*, *TDBrain*, *BeijingAir*, *BenzeneConcentration*, *AustraliaRainfall*, *KDDCup2018*, *PedestrianCounts*) representing a subset of the Unified Time Series Dataset (UTSD) [36].

This section provides a summary of descriptive statistics about the employed datasets, an example of two datasets with clear and unclear patterns, respectively, and dataset-specific preprocessing steps.

### B.1  Dataset statistics

In Tab. 6, we provide a comprehensive description of the datasets employed in this work and their corresponding statistics. Next, we describe in detail the methodology to derive such dataset statistics.

**Time steps and channels.** We counted the *number of time steps* and *number of channels*.

**Shannon entropy and spectral entropy.** Shannon entropy quantifies the average level of uncertainty or information content associated with the outcomes in a discrete variable $X$. Spectral entropy, a related concept, applies this measure to the frequency domain, using the normalized power spectral density as the probability distribution. Entropy is calculated as

$$H(X) = -\sum_{x \in \chi} p(x) \log p(x)$$

where $\chi$ is the set of all possible outcomes, and $p(x)$ is the probability of outcome $x$.

**Sample entropy.** Sample entropy is a statistical measure used to quantify the complexity or regularity of a time series. Unlike Shannon entropy, which evaluates uncertainty in discrete probability distributions, sample entropy assesses the likelihood that similar patterns in the time series remain similar at the next time step. It is defined as the negative natural logarithm of the conditional probability that two sequences of length $m$ that match within a tolerance $r$ will still match when extended to $m + 1$. A lower sample entropy indicates more regularity or predictability in the series, while a higher value suggests greater randomness or complexity. Sample entropy is calculated as

$$\text{SampEn}(m, r, N) = -\ln \frac{A}{B}$$

where $m$ is the embedding dimension, $r$ is the tolerance, $N$ is the total length of the time series, $A$ is the number of matching pairs of length $m + 1$, and $B$ is the number of matching pairs of length $m$.

**Stationarity.** In LTSF, stationarity is an important characteristic of time series, where the statistical characteristics, such as mean and variance, remain constant over time. We used the Augmented Dickey-Fuller (ADF) test to assess stationarity. This test evaluates the null hypothesis that a unit root is present and confirms stationarity if $\gamma < 0$ and the result is statistically significant. We report $\gamma$ since it scales with stationarity.

**Complexity.** In time series analysis, complexity refers to the irregularity or unpredictability in the data. We quantify complexity by using Higuchi's fractal dimension. A higher fractal dimension indicates greater complexity, while a lower value suggests more regularity or predictability in the data. Higuchi's fractal dimension is calculated as

$$D = \lim_{k \to 0} \frac{\log\left(\sum_{i=1}^{n}\left(\frac{|x_{i+k}-x_i|}{k}\right)\right)}{\log k}$$

where $x_i$ is a point, $k$ is the time scale, and the sum is taken over different segments of the time series.

**Inter-variate similarity.** As a proxy for inter-variate similarity, we provide the explained variance of the first principal component (PC1) obtained through principal component analysis (PCA). PC1 represents the direction of maximum variance in the data, capturing the dominant shared variation among the variables. The explained variance of PC1 quantifies the proportion of the total variance that is accounted for by this component. A higher explained variance indicates stronger similarity and shared dynamics among the variables, while a lower value suggests more independent behavior.

## B.2   Clear vs. unclear patterns

In Sec. 3, we assessed the performance of a linear model versus a transformer model on datasets with clear and unclear patterns, respectively. We use the same dataset as BasicTS+ [46] with an unclear pattern (*Exchange*) and replace their previously used PEMS with a clear pattern by *PedestrianCounts* (Fig. 5). The plots of the time series highlight contrasting characteristics: *Exchange* displays seemingly random trends, whereas *PedestrianCounts* exhibits evident cyclic behavior. To further emphasize this distinction, we provide butterfly plots, which reveal pronounced periodic patterns in *PedestrianCounts* and irregular, stochastic-like trends in *Exchange*. Additionally, the power spectrum analysis underscores this contrast, showing a dominant peak for *PedestrianCounts* and an absence of such peaks for *Exchange*.

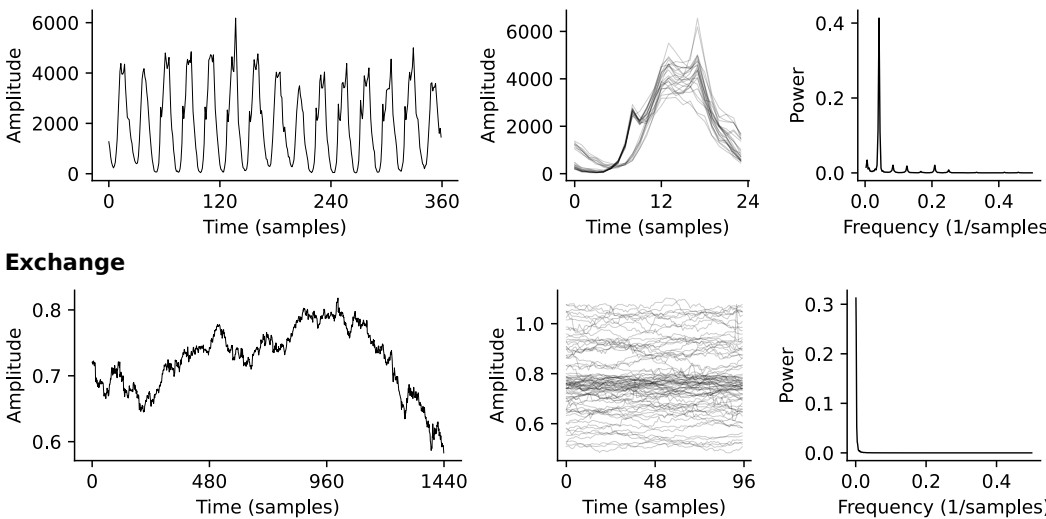

Figure 5: **Clear vs. unclear patterns (top vs bottom) in two datasets.**

## B.3   Preprocessing

We followed the preprocessing steps from TSlib [56]. Furthermore, we added functionality to import data from UTSD as curated by [36]. Since the UTSD datasets are magnitudes larger, we modified the respective dataloader to return sequences at a stride length $S = 100$ to accelerate the training.

## C   HP search details

**HP search.** To ensure a fair comparison, we performed an extensive HP search for all models on all the datasets. Specifically, we searched for an input length between 96 and 512, model size $d_m$ from 16 to 512, learning rate ranging from $10^{-5}$ to 0.1, and encoder layers between 1 and 5. The specific ranges are presented in Tab. 7. TimeMixer was limited to a maximum of 3 layers and a model size of 128 due to increasingly high memory demands ($> 49$GB of VRAM on an RTX A6000 GPU if $d_m > 128$ and $L = 720$). However, this is unlikely to affect performance, as the original HP search for all datasets in this study yielded results within these limits. We used the Optuna framework [1] with a budget of 40 trials to optimize the HPs. We employed the default `TPEsampler` for HP sampling and applied the `SuccessiveHalvingPruner`, configured with a minimum of three epochs and a reduction factor of two to prune unpromising trials. The search was conducted with a batch size of 8, a maximum of 15 epochs, and early stopping with a patience of 3 epochs. All models were optimized with the Adam optimizer and an exponentially decaying scheduler following the default TSLib configuration. The optimal HPs, determined by the minimum validation loss in the trials, were used to train and evaluate the final model across three random seeds to ensure robust results. We set the dimensions of the fully connected layers $d_f$ equal to $d_m$. The patch length, a parameter used in PatchTST and TimeXer (and by extension also iPatch), was set based on the characteristics of the datasets (Tab. 8). As visualized in Fig. 5, there are datasets with dominant periodic behavior. We set the patch length $P \approx \text{argmax}(\text{FFT}(X))$ wherever we observed such a natural pattern. Unsurprisingly, the patch length resulted in a span of one day in all datasets with a pattern. In the remaining cases, we set the patch length manually (*Exchange*, *MotorImagery*, *TDBrain*). We aligned with the idea of TimesNet, which introduced series segmentation based on dominant frequencies to enhance performance [60]. Moreover, the original works concluded that variations in patch length have minimal effects [39, 57]. For the rest of the model HPs, we default to the configurations provided in TSLib.

Table 7: **Searched HPs values.**

| Hyperparameter | TimeMixer | Other Models |
|---|---|---|
| Input Length | $\{96, 192, 336, 720\}$ | $\{96, 192, 336, 720\}$ |
| Learning Rate | $\{10^{-5}, 10^{-4}, 10^{-3}, 10^{-2}\}$ | $\{10^{-5}, 10^{-4}, 10^{-3}, 10^{-2}\}$ |
| #Layers | $\{1, 2, 3\}$ | $\{1, 2, 3, 4\}$ |
| $d_m$ | $\{16, 32, 64, 128\}$ | $\{16, 32, 64, 128, 256, 512\}$ |

Table 8: **Patch lengths.** Employed patch lengths for PatchTST, TimeXer, and iPatch models.

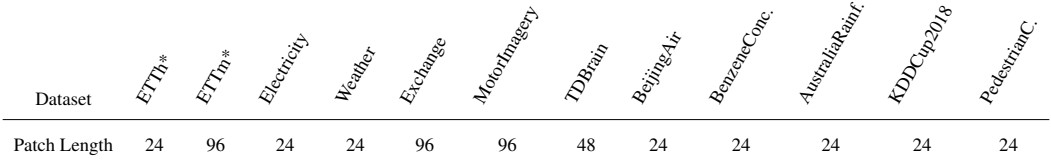

| Dataset | ETTh* | ETTm* | Electricity | Weather | Exchange | MotorImagery | TDBrain | BeijingAir | BenzeneConc. | AustraliaRainf. | KDDCup2018 | PedestrianC. |
|---|---|---|---|---|---|---|---|---|---|---|---|---|
| Patch Length | 24 | 96 | 24 | 24 | 96 | 96 | 48 | 24 | 24 | 24 | 24 | 24 |

## D   Implementation reliability

To assess the reliability of our implementation, we compare our results (best and average MSE) against the original works on popular datasets. Despite not being directly comparable for slight differences in setups, we align with previous values (Tab. 9).

## E   Stability of HP search

In this section, we perform a small proof-of-concept experiment to show the reliability of our HP search. For a subset of datasets and models, we performed two independent HP runs and consequent training of three models with the found optimal HPs to analyze the stability of the performed HP search. We focused on forecast horizon 96. In Fig. 6, we plot the MSE of the two searches over two

Table 9: **Implementation reliability.** To assess the reliability of our implementation, we compare our results (best and average MSE) against the original works on popular datasets. Despite not being directly comparable due to slight differences in experimental setups, we align with previous values.

| Model | Dataset | Original | Ours Min | Ours Mean |
|---|---|---|---|---|
| DLinear | ETT* | 0.370 | 0.378 | 0.378 |
| | Electricity | 0.166 | 0.162 | 0.162 |
| | Weather | 0.246 | 0.244 | 0.244 |
| PatchTST | ETT* | 0.338 | 0.332 | 0.336 |
| | Electricity | 0.159 | 0.163 | 0.164 |
| | Weather | 0.225 | 0.224 | 0.225 |
| iTransformer | ETT* | 0.383 | 0.342 | 0.344 |
| | Electricity | 0.178 | 0.162 | 0.166 |
| | Weather | 0.258 | 0.237 | 0.239 |
| TimeMixer | ETT* | 0.333 | 0.342 | 0.349 |
| | Electricity | 0.156 | 0.154 | 0.156 |
| | Weather | 0.241 | 0.225 | 0.231 |
| TimeXer | ETT* | 0.363 | 0.341 | 0.346 |
| | Electricity | 0.171 | 0.165 | 0.170 |
| | Weather | 0.241 | 0.222 | 0.224 |

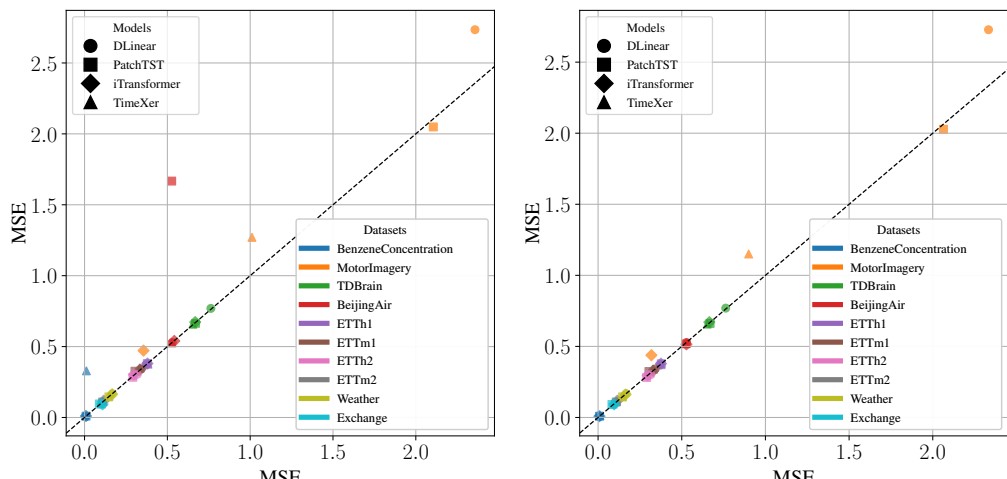

Figure 6: **HP search variability.** Comparison among two independent HP search results in terms of MSE for forecast horizon 96. The average of the final three models shows minimal variability except for a few cases (left), while the best model is even more stable (right).

opposing axes, meaning that points on the diagonal indicate a very stable search that leads to the same final result. To provide an even more comprehensive analysis, we show the average of the three final models (left) and also the best model out of the final three (right). The averages show a few cases with slight variability, namely PatchTST on *BejingAir* and TimeXer on *BenzeneConcentration*, but overall, the experiment proves a reliable and stable HP search across independent runs. In the case of best results (Fig. 6, right), the variability is even less noticeable.

# F Statistical tests

## F.1 Friedman test

The Friedman test [17, 18] is a non-parametric statistical method designed as an alternative to repeated-measures ANOVA. It enables the comparison of multiple algorithms across multiple datasets when the assumptions of parametric tests may not hold. The test works by ranking the algorithms on

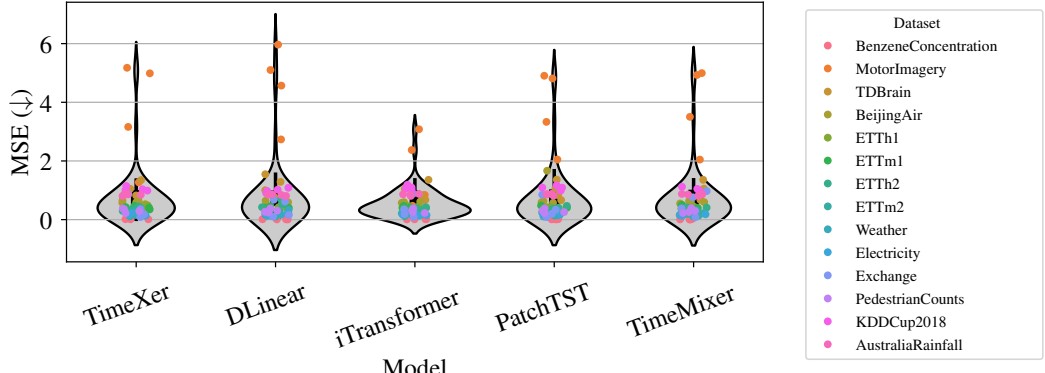

Figure 7: **Violin plot for datasets and horizons.** We provide an alternative view of model performance on datasets and prediction horizons. Each point represents the average MSE obtained for a given forecast horizon over three independent seeds with the found HP configuration. It is clearly visible that the difference in average scores solely depends on the *MotorImagery* dataset since baseline scores are comparable over the rest of the dataset-horizon setups.

each dataset separately, with the best-performing algorithm assigned a rank of 1, the second-best a rank of 2, and so forth. In cases of ties, average ranks are assigned across the tied algorithms.

Let $r_i^j$ denote the rank of the $j$-th algorithm out of $k$ algorithms on the $i$-th dataset out of $N$ datasets. The Friedman test evaluates the average ranks of the algorithms, calculated as $R^j = \frac{1}{N} \sum_{i=1}^{N} r_i^j$. Under the null hypothesis, which assumes that all algorithms are equivalent in performance and thus their ranks $R^j$ should be approximately equal, the Friedman statistic is given by:

$$\chi_F^2 = \frac{12N}{k(k+1)} \left[ \sum_{j=1}^{k} (R^j)^2 - \frac{k(k+1)^2}{4} \right]$$

For sufficiently large values of $N$ and $k$, as a rule of thumb $N > 10$ and $k > 5$ [13], this statistic follows a $\chi^2$ distribution with $k - 1$ degrees of freedom. Note our experimental setup aligns with these conditions.

The Friedman test, though less powerful than parametric repeated-measures ANOVA when its assumptions are met, is more robust in handling violations of these assumptions, with [18] observing largely consistent results between the two tests across 56 independent problems.

### F.2   Sign test

The sign test [43] is a non-parametric statistical method commonly used to compare the performance of two algorithms across multiple datasets. It operates by evaluating the number of datasets on which each algorithm outperforms the other, assuming that the outcomes are independent and identically distributed. Contrary to popular belief, counting only significant wins and losses actually makes the tests less reliable, as it imposes an arbitrary threshold of $p < 0.05$ to determine meaningfulness [13].

Under the null hypothesis, it is assumed that the two algorithms are equivalent in their performance, and thus, each algorithm has an equal probability (0.5) of outperforming the other on a given dataset. This leads to the number of wins for either algorithm following a binomial distribution with parameters $N$ (the total number of datasets) and $p = 0.5$. The null hypothesis is rejected if the observed number of wins for one algorithm is significantly different from $\frac{N}{2}$, indicating that one algorithm systematically outperforms the other.

For small sample sizes, as in our case, with a total number of datasets being equal to 14, critical values can be determined directly from the cumulative distribution function of the binomial distribution. For larger sample sizes, the central limit theorem allows for an approximation using the normal distribution. Specifically, the number of wins under the null hypothesis can be approximated by a

normal distribution with mean $\mu = N/2$ and standard deviation $\sigma = \sqrt{N}/2$. In cases where there are ties in performance, these ties are treated as supporting evidence for the null hypothesis. To account for this, ties are split evenly between the two algorithms. If the number of tied datasets is odd, one tie is disregarded to ensure that only whole numbers are assigned to each algorithm.

## G  Baseline comparisons

We provide an overview of included baselines for the top-performing LTSF models (Tab. 10). We observe that previous models were replaced by recent ones as the field progressed. We highlight that TimeMixer was not included as a baseline in TimeXer, although it was available at the time.

Table 10: **Included baseline models from top-performing models in long-term forecasting**. x: included; **o**: introduced

| Model | DLinear [66] | PatchTST [39] | TimeMixer [55] | iTransformer [34] | TimeXer [57] | FEDformer [69] | Autoformer [61] | Informer [68] | Pyraformer [33] | LogTrans [30] | Stationary [35] | Crossformer [67] | TimesNet [60] | SCINet [32] | Rlinear [31] | TiDE [12] | others |
|---|---|---|---|---|---|---|---|---|---|---|---|---|---|---|---|---|---|
| DLinear [66] | **o** | | | | | x | x | x | x | x | | | | | | | |
| PatchTST [39] | x | **o** | | | | x | x | x | x | x | | | | | | | |
| TimeMixer [55] | x | x | **o** | | | x | x | x | | | x | x | x | | | | x |
| iTransformer [34] | x | x | | **o** | | x | x | | | | x | x | x | x | x | x | |
| TimeXer [57] | x | x | | x | **o** | x | | | x | | x | x | x | x | x | x | |

## H  Full results

Tab. 12 and Tab. 13 shows the full results from our extensive experiments. We present the MSE and MAE for all forecast horizons $T \in \{96, 192, 336, 720\}$ and their average, respectively. To provide a comprehensive view, we show the average (Tab. 12) and the best (Tab. 13) values over three random seeds. Tab. 14 presents the corresponding full results for iPatch.

For completeness, we also included two simple baselines to contextualize model performance in the challenging task of long-term forecasting. We included ARIMA as a classic statistical forecasting model [5] and Last Observation Carried Forward (LOCF), which predict all future time steps by repeating the last observed value from the input sequence [22]. We observe that, on average, the more recent machine learning models perform substantially better than the simple baselines (Tab. 12, Tab. 13). However, ARIMA and LOCF are among the best-performing models on *Exchange*. This is not surprising, given that stock market data lacks obvious periodicities and was previously shown to be best predicted by simple baselines [22]. This further supports our broader message: no model is consistently best, and performance can vary widely depending on the dataset.

## I  Efficiency comparisons

**Efficiency metrics setup.** The efficiency metrics were computed by evaluating model performance across 1,000 iterations using synthetic input and target data shaped according to the specified sequence length and prediction horizon stemming from the optimal experimental setup found during the HP search. During each iteration, the model was executed in either training or inference mode with a batch size of one, depending on the configuration, and both the throughput (TP) and peak GPU memory (`memory`) usage were recorded. The TP was quantified as the number of processed sequences per second, calculated by dividing the total number of sequences by the elapsed time. We run the above analyses on a machine equipped with an RTX 3060 NVIDIA GPU and an 11th Gen Intel(R) Core(TM) i7-11700F CPU. Additionally, we computed the number of floating-point operations (`FLOPs`) and total trainable parameters (`#params`), offering insight into the theoretical computational complexity of evaluated models. We subsequently scaled all efficiency metrics for interpretability: parameters to millions, FLOPs to gigaflops, and memory usage to megabytes (MB).

Table 11: **Efficiency metrics.** Average efficiency metrics for all optimized models over datasets and prediction horizons (1,000 iterations, batch size of 1). **Best** and second-best are highlighted.

| Metric | DLinear | PatchTST | iTransformer | TimeMixer | TimeXer |
|---|---|---|---|---|---|
| Train TP ($\frac{\text{seq}}{\text{s}}$) | **1,497.39** | 240.06 | 617.98 | 74.14 | 143.51 |
| Train memory (MB) | **132.17** | 184.23 | 142.12 | 373.30 | 214.23 |
| Test TP ($\frac{\text{seq}}{\text{s}}$) | **4,284.35** | 761.51 | 931.86 | 243.74 | 464.77 |
| Test memory (MB) | **129.64** | 153.88 | 136.85 | 192.27 | 155.63 |
| FLOPS (G) | **0.02** | 0.69 | 0.20 | 15.19 | 4.40 |
| #params (M) | **0.36** | 2.29 | 1.56 | 2.50 | 3.91 |

**Efficiency metrics results.** In Tab. 11, we provide an overview of the average efficiency metrics for all models over datasets and prediction horizons. Note that for each dataset-horizon combination, we compute the metrics for the configuration found with the HP search. Unsurprisingly, DLinear consistently demonstrates superior efficiency across all metrics, achieving the highest training and inference throughput, lowest memory usage, minimal FLOPS, and the smallest parameter count. iTransformer, followed by PatchTST, is the second most efficient model in all metrics, showing an advantageous trade-off between throughput and memory across the more complex models. TimeMixer and TimeXer lag particularly behind in terms of efficiency against DLinear. Specifically, DLinear attains a training throughput of $\approx$1,500 sequences/s, making it roughly $10\times$ faster than TimeXer and over $20\times$ faster than TimeMixer. In terms of training memory, DLinear requires on average 132.17 MB, which is about $2.8\times$ less than TimeMixer and $1.6\times$ less than TimeXer.

Table 12: **Full results.** Mean values for all prediction lengths. **Best** and second-best are highlighted.

| Dataset | Metric | DLinear MAE | DLinear MSE | PatchTST MAE | PatchTST MSE | iTransformer MAE | iTransformer MSE | TimeMixer MAE | TimeMixer MSE | TimeXer MAE | TimeXer MSE | ARIMA MAE | ARIMA MSE | CLOF MAE | CLOF MSE |
|---|---|---|---|---|---|---|---|---|---|---|---|---|---|---|---|
| ETTh1 | H96 | 0.3938 | 0.3703 | 0.4044 | 0.3798 | 0.4031 | 0.3806 | 0.3967 | 0.3702 | 0.4001 | 0.3758 | 0.5976 | 0.9436 | 0.7132 | 1.2944 |
|  | H192 | 0.4366 | 0.4225 | 0.4277 | 0.4176 | 0.4384 | 0.4133 | 0.4349 | 0.411 | 0.4297 | 0.4171 | 0.6157 | 0.9675 | 0.7331 | 1.3249 |
|  | H336 | 0.532 | 0.5427 | 0.4329 | 0.4221 | 0.4423 | 0.4351 | 0.4416 | 0.438 | 0.4345 | 0.4247 | 0.6295 | 0.9741 | 0.746 | 1.3299 |
|  | H720 | 0.5464 | 0.5608 | 0.4626 | 0.4378 | 0.4876 | 0.4684 | 0.4985 | 0.4968 | 0.4957 | 0.4831 | 0.6414 | 0.9746 | 0.755 | 1.3351 |
|  | Average | 0.4772 | 0.4741 | 0.4319 | 0.4143 | 0.4428 | 0.4244 | 0.4429 | 0.429 | 0.44 | 0.4252 | 0.6211 | 0.9649 | 0.7368 | 1.3211 |
| ETTm1 | H96 | 0.3736 | 0.3415 | 0.3737 | 0.3263 | 0.387 | 0.3374 | 0.3869 | 0.341 | 0.3961 | 0.3482 | 0.6376 | 0.8181 | 0.7771 | 1.4631 |
|  | H192 | 0.407 | 0.4015 | 0.412 | 0.3618 | 0.4254 | 0.4028 | 0.4272 | 0.3972 | 0.4279 | 0.401 | 0.6804 | 0.9119 | 0.8046 | 1.5172 |
|  | H336 | 0.4349 | 0.4116 | 0.4445 | 0.4062 | 0.4519 | 0.4232 | 0.4808 | 0.4976 | 0.4612 | 0.4277 | 0.7135 | 0.983 | 0.8313 | 1.5767 |
|  | H720 | 0.4716 | 0.4578 | 0.4955 | 0.4799 | 0.485 | 0.4687 | 0.4997 | 0.4904 | 0.4931 | 0.4715 | 0.7524 | 1.0647 | 0.8666 | 1.6467 |
|  | Average | 0.4218 | 0.4031 | 0.4314 | 0.3936 | 0.4373 | 0.408 | 0.4487 | 0.4315 | 0.4446 | 0.4121 | 0.696 | 0.9444 | 0.8199 | 1.5509 |
| ETTh2 | H96 | 0.3677 | 0.3042 | 0.3414 | 0.2833 | 0.3515 | 0.3026 | 0.3439 | 0.287 | 0.3489 | 0.2965 | 0.3875 | 0.3658 | 0.4216 | 0.4317 |
|  | H192 | 0.4429 | 0.4232 | 0.3919 | 0.3602 | 0.4016 | 0.3837 | 0.4055 | 0.3876 | 0.3971 | 0.3658 | 0.4421 | 0.4628 | 0.4725 | 0.5337 |
|  | H336 | 0.4875 | 0.5127 | 0.4263 | 0.406 | 0.4248 | 0.403 | 0.4288 | 0.4144 | 0.4266 | 0.4016 | 0.487 | 0.5282 | 0.5109 | 0.5973 |
|  | H720 | 0.599 | 0.6997 | 0.48 | 0.4677 | 0.4436 | 0.4245 | 0.4395 | 0.4086 | 0.4524 | 0.4441 | 0.5238 | 0.5755 | 0.519 | 0.5945 |
|  | Average | 0.4743 | 0.4849 | 0.4099 | 0.3793 | 0.4054 | 0.3784 | 0.4044 | 0.3744 | 0.4063 | 0.377 | 0.4601 | 0.4831 | 0.481 | 0.5393 |
| ETTm2 | H96 | 0.2215 | 0.1081 | 0.2243 | 0.1093 | 0.2325 | 0.1179 | 0.2253 | 0.1102 | 0.2275 | 0.1136 | 0.2978 | 0.1922 | 0.3099 | 0.2018 |
|  | H192 | 0.2471 | 0.134 | 0.2561 | 0.1396 | 0.2617 | 0.1483 | 0.2532 | 0.1399 | 0.2689 | 0.1562 | 0.3237 | 0.2241 | 0.3366 | 0.2353 |
|  | H336 | 0.2713 | 0.1616 | 0.2811 | 0.1675 | 0.2899 | 0.1826 | 0.2848 | 0.1792 | 0.287 | 0.1785 | 0.3484 | 0.2583 | 0.3613 | 0.2704 |
|  | H720 | 0.3044 | 0.2008 | 0.3148 | 0.2089 | 0.3195 | 0.2157 | 0.3183 | 0.2179 | 0.3235 | 0.2246 | 0.3889 | 0.3239 | 0.4009 | 0.3348 |
|  | Average | 0.2611 | 0.1511 | 0.2691 | 0.1563 | 0.2759 | 0.1661 | 0.2704 | 0.1618 | 0.2767 | 0.1682 | 0.3397 | 0.2496 | 0.3522 | 0.2606 |
| Electricity | H96 | 0.2298 | 0.133 | 0.2274 | 0.1301 | 0.23 | 0.135 | 0.228 | 0.1311 | 0.2398 | 0.1375 | 0.8619 | 1.3055 | 0.9455 | 1.5878 |
|  | H192 | 0.243 | 0.1472 | 0.2415 | 0.1466 | 0.2527 | 0.1527 | 0.2429 | 0.1466 | 0.2485 | 0.1487 | 0.9 | 1.4572 | 0.9507 | 1.5957 |
|  | H336 | 0.2621 | 0.1633 | 0.2718 | 0.1725 | 0.2686 | 0.1696 | 0.2645 | 0.1653 | 0.2611 | 0.1626 | 0.954 | 1.824 | 0.9614 | 1.6182 |
|  | H720 | 0.2991 | 0.2031 | 0.3045 | 0.207 | 0.2975 | 0.2046 | 0.2811 | 0.1826 | 0.321 | 0.2293 | 1.0879 | 3.6237 | 0.9754 | 1.6467 |
|  | Average | 0.2585 | 0.1616 | 0.2613 | 0.164 | 0.2622 | 0.1655 | 0.2541 | 0.1564 | 0.2676 | 0.1695 | 0.9509 | 2.0526 | 0.9582 | 1.6121 |
| Weather | H96 | 0.2262 | 0.1678 | 0.1952 | 0.1455 | 0.2158 | 0.164 | 0.2114 | 0.1567 | 0.1991 | 0.1472 | 0.2524 | 0.2118 | 0.2542 | 0.2591 |
|  | H192 | 0.275 | 0.2156 | 0.2465 | 0.1972 | 0.2582 | 0.2076 | 0.2545 | 0.2044 | 0.2415 | 0.192 | 0.2935 | 0.2548 | 0.2917 | 0.3092 |
|  | H336 | 0.3158 | 0.2625 | 0.2817 | 0.2431 | 0.2999 | 0.2601 | 0.2888 | 0.2535 | 0.2858 | 0.2451 | 0.3403 | 0.3204 | 0.3377 | 0.3764 |
|  | H720 | 0.3767 | 0.3307 | 0.3298 | 0.3123 | 0.3427 | 0.322 | 0.3284 | 0.3108 | 0.3291 | 0.3103 | 0.3963 | 0.4156 | 0.3935 | 0.4652 |
|  | Average | 0.2984 | 0.2442 | 0.2633 | 0.2245 | 0.2791 | 0.2385 | 0.2707 | 0.2313 | 0.2639 | 0.2237 | 0.3206 | 0.3007 | 0.3193 | 0.3525 |
| Exchange | H96 | 0.2295 | 0.1024 | 0.2032 | 0.0861 | 0.2142 | 0.092 | 0.2014 | 0.0838 | 0.2052 | 0.0865 | 0.1986 | 0.0823 | 0.1964 | 0.0811 |
|  | H192 | 0.2901 | 0.1581 | 0.3174 | 0.1975 | 0.3075 | 0.1839 | 0.5262 | 0.7602 | 0.3021 | 0.1805 | 0.2889 | 0.1678 | 0.2887 | 0.1671 |
|  | H336 | 0.5813 | 0.6157 | 0.4235 | 0.3383 | 0.4309 | 0.3545 | 0.4485 | 0.3844 | 0.4328 | 0.3594 | 0.4058 | 0.3163 | 0.3978 | 0.3057 |
|  | H720 | 0.6363 | 0.6783 | 0.6928 | 0.8487 | 0.71 | 0.8839 | 0.7296 | 0.9732 | 0.7067 | 0.8682 | 0.6872 | 0.8138 | 0.6764 | 0.8101 |
|  | Average | 0.4343 | 0.3886 | 0.4092 | 0.3676 | 0.4156 | 0.3786 | 0.4764 | 0.5504 | 0.4117 | 0.3737 | 0.3952 | 0.3451 | 0.3898 | 0.341 |
| MotorImagery | H96 | 0.9128 | 2.7345 | 0.7809 | 2.0489 | 0.1654 | 0.4718 | 0.754 | 2.0469 | 0.3444 | 1.2702 | 1.1658 | 4.7017 | 0.8701 | 2.1921 |
|  | H192 | 1.1518 | 4.5688 | 0.9406 | 3.3327 | 0.2719 | 0.8407 | 0.9481 | 3.5042 | 0.8772 | 3.1613 | 1.3779 | 6.4838 | 1.04 | 3.4885 |
|  | H336 | 1.3276 | 5.9655 | 1.0953 | 4.8131 | 0.4366 | 2.3733 | 1.1189 | 4.995 | 1.1727 | 5.1769 | 1.4519 | 7.0403 | 1.2109 | 5.0798 |
|  | H720 | 1.3412 | 5.1007 | 1.2071 | 4.9048 | 0.66 | 3.0805 | 1.2243 | 4.9304 | 1.2676 | 4.9876 | 1.2894 | 5.047 | 1.3409 | 5.305 |
|  | Average | 1.1834 | 4.5924 | 1.006 | 3.7749 | 0.3835 | 1.6916 | 1.0113 | 3.8691 | 0.9155 | 3.649 | 1.3213 | 5.8182 | 1.1155 | 4.0164 |
| TDBrain | H96 | 0.65 | 0.7693 | 0.5958 | 0.6641 | 0.5979 | 0.6726 | 0.5959 | 0.673 | 0.5911 | 0.6574 | 0.675 | 0.8627 | 0.764 | 1.106 |
|  | H192 | 0.7463 | 1.0022 | 0.6695 | 0.8348 | 0.6671 | 0.8314 | 0.6688 | 0.8374 | 0.6633 | 0.824 | 0.7387 | 1.0133 | 0.8257 | 1.2615 |
|  | H336 | 0.846 | 1.2863 | 0.7543 | 1.066 | 0.7488 | 1.0531 | 0.7498 | 1.0568 | 0.7482 | 1.0505 | 0.809 | 1.2072 | 0.8951 | 1.4667 |
|  | H720 | 0.9643 | 1.5473 | 0.8798 | 1.3633 | 0.8757 | 1.3561 | 0.8767 | 1.3575 | 0.8722 | 1.3479 | 0.9204 | 1.495 | 1.0072 | 1.7817 |
|  | Average | 0.8016 | 1.1513 | 0.7248 | 0.9821 | 0.7224 | 0.9783 | 0.7228 | 0.9812 | 0.7187 | 0.97 | 0.7858 | 1.1445 | 0.873 | 1.404 |
| BeijingAir | H96 | 0.4409 | 0.529 | 0.4302 | 0.5221 | 0.4415 | 0.5399 | 0.4516 | 0.5515 | 0.4321 | 0.5261 | 0.5041 | 0.6652 | 0.5983 | 1.1597 |
|  | H192 | 0.4635 | 0.569 | 0.4543 | 0.5723 | 0.4584 | 0.571 | 0.4717 | 0.5955 | 0.4538 | 0.5737 | 0.5406 | 0.7403 | 0.6461 | 1.214 |
|  | H336 | 0.4732 | 0.5912 | 0.466 | 0.5908 | 0.4682 | 0.5948 | 0.4667 | 0.5889 | 0.4652 | 0.5881 | 0.5477 | 0.7559 | 0.6281 | 1.077 |
|  | H720 | 0.5088 | 0.6426 | 0.4879 | 0.6272 | 0.4883 | 0.6147 | 0.4948 | 0.6309 | 0.4961 | 0.6393 | 0.5706 | 0.7879 | 0.6531 | 1.1493 |
|  | Average | 0.4716 | 0.5829 | 0.4596 | 0.5781 | 0.4641 | 0.5801 | 0.4712 | 0.5917 | 0.4618 | 0.5818 | 0.5408 | 0.7373 | 0.6314 | 1.15 |
| BenzeneConcentration | H96 | 0.0162 | 0.0067 | 0.0407 | 0.0095 | 0.0582 | 0.0105 | 0.0163 | 0.0059 | 0.0405 | 0.0095 | 0.7636 | 0.9311 | 0.8622 | 1.2643 |
|  | H192 | 0.0148 | 0.0066 | 0.027 | 0.0097 | 0.0367 | 0.0077 | 0.0185 | 0.0063 | 0.0374 | 0.0103 | 0.81 | 1.0032 | 0.9001 | 1.3454 |
|  | H336 | 0.0192 | 0.0082 | 0.0415 | 0.0094 | 0.0586 | 0.0133 | 0.0285 | 0.0079 | 0.0321 | 0.0114 | 0.8294 | 1.0425 | 0.8703 | 1.2813 |
|  | H720 | 0.0282 | 0.0115 | 0.0583 | 0.0157 | 0.0581 | 0.0153 | 0.0266 | 0.0126 | 0.0376 | 0.0156 | 0.8564 | 1.104 | 0.8705 | 1.281 |
|  | Average | 0.0196 | 0.0082 | 0.0419 | 0.0111 | 0.0529 | 0.0117 | 0.0225 | 0.0082 | 0.0369 | 0.0117 | 0.8149 | 1.0202 | 0.8758 | 1.293 |
| AustraliaRainfall | H96 | 0.7298 | 0.8063 | 0.7317 | 0.8154 | 0.7271 | 0.809 | 0.7286 | 0.8135 | 0.7277 | 0.8075 | 1.0919 | 1.9107 | 1.0833 | 1.8775 |
|  | H192 | 0.7508 | 0.8388 | 0.7583 | 0.8623 | 0.7548 | 0.8507 | 0.7555 | 0.8546 | 0.7527 | 0.8475 | 1.1221 | 1.9776 | 1.1147 | 1.9491 |
|  | H336 | 0.76 | 0.8502 | 0.7641 | 0.8644 | 0.7631 | 0.8633 | 0.7647 | 0.8678 | 0.762 | 0.8614 | 1.1276 | 1.9762 | 1.1191 | 1.9442 |
|  | H720 | 0.7645 | 0.8576 | 0.7721 | 0.8773 | 0.7701 | 0.8743 | 0.7707 | 0.8761 | 0.7689 | 0.8709 | 1.1406 | 2.0112 | 1.1319 | 1.9792 |
|  | Average | 0.7513 | 0.8382 | 0.7566 | 0.8548 | 0.7537 | 0.8493 | 0.7548 | 0.853 | 0.7528 | 0.8468 | 1.1206 | 1.9689 | 1.1123 | 1.9375 |
| KDDCup2018 | H96 | 0.6273 | 1.0863 | 0.6524 | 1.1694 | 0.6547 | 1.1861 | 0.6301 | 1.1202 | 0.6558 | 1.1455 | 0.6611 | 1.1519 | 0.7816 | 1.5059 |
|  | H192 | 0.6266 | 0.9782 | 0.6452 | 1.0903 | 0.651 | 1.0941 | 0.6407 | 1.0484 | 0.6289 | 1.0342 | 0.6584 | 1.0689 | 0.7744 | 1.4141 |
|  | H336 | 0.6482 | 1.0208 | 0.6576 | 1.0924 | 0.6533 | 1.0649 | 0.6344 | 1.0111 | 0.6377 | 1.0101 | 0.654 | 1.0403 | 0.7824 | 2.0965 |
|  | H720 | 0.6185 | 0.9014 | 0.6342 | 0.9905 | 0.635 | 1.0076 | 0.6188 | 0.9589 | 0.6359 | 0.9901 | 0.6812 | 1.0882 | 0.8212 | 1.4917 |
|  | Average | 0.6302 | 0.9967 | 0.6474 | 1.0856 | 0.6485 | 1.0882 | 0.631 | 1.0346 | 0.6396 | 1.045 | 0.6637 | 1.0873 | 0.7899 | 1.6271 |
| PedestrianCounts | H96 | 0.2578 | 0.2388 | 0.2541 | 0.2203 | 0.2463 | 0.2258 | 0.2482 | 0.2246 | 0.2449 | 0.219 | 1.0932 | 2.6555 | 0.9544 | 1.9948 |
|  | H192 | 0.272 | 0.2659 | 0.2742 | 0.2565 | 0.2553 | 0.255 | 0.2673 | 0.2586 | 0.3674 | 0.3771 | 1.0999 | 2.6442 | 0.9645 | 2.029 |
|  | H336 | 0.295 | 0.3072 | 0.3026 | 0.3046 | 0.3057 | 0.3122 | 0.2844 | 0.2973 | 0.2871 | 0.2886 | 1.123 | 2.6634 | 0.9879 | 2.0655 |
|  | H720 | 0.3322 | 0.3815 | 0.3407 | 0.3844 | 0.3341 | 0.3859 | 0.3381 | 0.3888 | 0.3271 | 0.3601 | 1.2082 | 3.0112 | 1.0517 | 2.2378 |
|  | Average | 0.2892 | 0.2983 | 0.2929 | 0.2915 | 0.2854 | 0.2947 | 0.2845 | 0.2923 | 0.3066 | 0.3112 | 1.1311 | 2.7436 | 0.9896 | 2.0818 |
| Average | Average | 0.4837 | 0.7697 | 0.4575 | 0.6913 | 0.4163 | 0.5467 | 0.4618 | 0.7118 | 0.453 | 0.6832 | 0.7258 | 1.4186 | 0.746 | 1.3919 |
| Rank | Average | 3.5 | 3.64 | 2.93 | 2.64 | 3.29 | 3.36 | 3.14 | 3.21 | 3.07 | 3.14 | 5.86 | 5.79 | 6.21 | 6.21 |

Table 13: **Full results.** Best values for all prediction lengths. **Best** and second-best are highlighted.

| Dataset | Model | DLinear | | PatchTST | | iTransformer | | TimeMixer | | TimeXer | | ARIMA | | CLOF | |
|---|---|---|---|---|---|---|---|---|---|---|---|---|---|---|---|
| | Metric | MAE | MSE | MAE | MSE | MAE | MSE | MAE | MSE | MAE | MSE | MAE | MSE | MAE | MSE |
| ETTh1 | H96 | 0.3932 | 0.3698 | 0.4034 | 0.3779 | 0.4002 | 0.3787 | 0.3958 | 0.3682 | 0.3976 | 0.3712 | 0.5976 | 0.9436 | 0.7132 | 1.2944 |
| | H192 | 0.4366 | 0.4225 | 0.4231 | 0.4125 | 0.437 | 0.4109 | 0.4261 | 0.4045 | 0.4225 | 0.4075 | 0.6157 | 0.9675 | 0.7331 | 1.3249 |
| | H336 | 0.532 | 0.5426 | 0.4306 | 0.417 | 0.4413 | 0.4327 | 0.4381 | 0.4343 | 0.4308 | 0.4217 | 0.6295 | 0.9741 | 0.746 | 1.3299 |
| | H720 | 0.5428 | 0.5551 | 0.4543 | 0.4231 | 0.4847 | 0.4646 | 0.4944 | 0.4909 | 0.4934 | 0.4812 | 0.6414 | 0.9746 | 0.755 | 1.3351 |
| | Average | 0.4761 | 0.4725 | 0.4279 | 0.4076 | 0.4408 | 0.4217 | 0.4386 | 0.4245 | 0.4361 | 0.4204 | 0.6211 | 0.9649 | 0.7368 | 1.3211 |
| ETTm1 | H96 | 0.3736 | 0.3415 | 0.3705 | 0.3236 | 0.3801 | 0.3294 | 0.3777 | 0.3292 | 0.3902 | 0.3393 | 0.6376 | 0.8181 | 0.7771 | 1.4631 |
| | H192 | 0.407 | 0.4015 | 0.4097 | 0.3613 | 0.4243 | 0.4008 | 0.4126 | 0.3692 | 0.422 | 0.3934 | 0.6804 | 0.9119 | 0.8046 | 1.5172 |
| | H336 | 0.4325 | 0.4093 | 0.4431 | 0.4019 | 0.4469 | 0.4157 | 0.4733 | 0.487 | 0.4575 | 0.4246 | 0.7135 | 0.983 | 0.8313 | 1.5767 |
| | H720 | 0.4701 | 0.4567 | 0.4937 | 0.4792 | 0.482 | 0.4657 | 0.4905 | 0.4773 | 0.4894 | 0.4645 | 0.7524 | 1.0647 | 0.8666 | 1.6467 |
| | Average | 0.4208 | 0.4023 | 0.4292 | 0.3915 | 0.4333 | 0.4029 | 0.4385 | 0.4157 | 0.4398 | 0.4054 | 0.696 | 0.9444 | 0.8199 | 1.5509 |
| ETTh2 | H96 | 0.3655 | 0.3014 | 0.3389 | 0.2807 | 0.348 | 0.2964 | 0.3413 | 0.2836 | 0.346 | 0.2948 | 0.3875 | 0.3658 | 0.4216 | 0.4317 |
| | H192 | 0.4428 | 0.4231 | 0.3902 | 0.3563 | 0.3972 | 0.379 | 0.4041 | 0.3843 | 0.3954 | 0.3624 | 0.4421 | 0.4628 | 0.4725 | 0.5337 |
| | H336 | 0.4874 | 0.5126 | 0.4243 | 0.4026 | 0.4226 | 0.3977 | 0.4201 | 0.3975 | 0.4225 | 0.3977 | 0.487 | 0.5282 | 0.5109 | 0.5973 |
| | H720 | 0.5988 | 0.6991 | 0.4715 | 0.4548 | 0.4432 | 0.4237 | 0.4388 | 0.4081 | 0.4515 | 0.4435 | 0.5238 | 0.5755 | 0.519 | 0.5945 |
| | Average | 0.4736 | 0.4841 | 0.4062 | 0.3736 | 0.4028 | 0.3742 | 0.4011 | 0.3684 | 0.4038 | 0.3746 | 0.4601 | 0.4831 | 0.481 | 0.5393 |
| ETTm2 | H96 | 0.2215 | 0.1081 | 0.2234 | 0.1089 | 0.2322 | 0.1176 | 0.224 | 0.1089 | 0.2273 | 0.1131 | 0.2978 | 0.1922 | 0.3099 | 0.2018 |
| | H192 | 0.247 | 0.1339 | 0.2547 | 0.1384 | 0.2608 | 0.1473 | 0.2527 | 0.1395 | 0.2636 | 0.1505 | 0.3237 | 0.2241 | 0.3366 | 0.2353 |
| | H336 | 0.2712 | 0.1616 | 0.279 | 0.1658 | 0.2891 | 0.1816 | 0.2826 | 0.1746 | 0.2853 | 0.1771 | 0.3484 | 0.2583 | 0.3613 | 0.2704 |
| | H720 | 0.3044 | 0.2008 | 0.3133 | 0.2079 | 0.3175 | 0.2135 | 0.315 | 0.2161 | 0.3213 | 0.2211 | 0.3889 | 0.3239 | 0.4009 | 0.3348 |
| | Average | 0.261 | 0.1511 | 0.2676 | 0.1552 | 0.2749 | 0.165 | 0.2686 | 0.1598 | 0.2744 | 0.1654 | 0.3397 | 0.2496 | 0.3522 | 0.2606 |
| Electricity | H96 | 0.2298 | 0.133 | 0.227 | 0.13 | 0.229 | 0.134 | 0.227 | 0.1299 | 0.2393 | 0.1372 | 0.8619 | 1.3055 | 0.9455 | 1.5878 |
| | H192 | 0.243 | 0.1472 | 0.2412 | 0.1463 | 0.2459 | 0.1488 | 0.2423 | 0.1459 | 0.2466 | 0.1476 | 0.9 | 1.4572 | 0.9507 | 1.5957 |
| | H336 | 0.2612 | 0.1627 | 0.2713 | 0.171 | 0.2602 | 0.1629 | 0.2618 | 0.1628 | 0.2575 | 0.1599 | 0.954 | 1.824 | 0.9614 | 1.6182 |
| | H720 | 0.2991 | 0.2031 | 0.3038 | 0.2062 | 0.2957 | 0.2024 | 0.2785 | 0.1785 | 0.3106 | 0.2135 | 1.0879 | 3.6237 | 0.9753 | 1.6467 |
| | Average | 0.2583 | 0.1615 | 0.2608 | 0.1634 | 0.2577 | 0.162 | 0.2524 | 0.1543 | 0.2635 | 0.1645 | 0.9509 | 2.0526 | 0.9582 | 1.6121 |
| Weather | H96 | 0.226 | 0.1678 | 0.1947 | 0.1451 | 0.2141 | 0.1616 | 0.2012 | 0.1487 | 0.1975 | 0.1459 | 0.2524 | 0.2118 | 0.2542 | 0.2591 |
| | H192 | 0.2749 | 0.2156 | 0.2458 | 0.1962 | 0.2579 | 0.2071 | 0.2437 | 0.1938 | 0.2392 | 0.1892 | 0.2935 | 0.2548 | 0.2917 | 0.3092 |
| | H336 | 0.3157 | 0.2625 | 0.2806 | 0.2415 | 0.2987 | 0.258 | 0.2849 | 0.2496 | 0.2817 | 0.2426 | 0.3403 | 0.3204 | 0.3377 | 0.3764 |
| | H720 | 0.3767 | 0.3307 | 0.3279 | 0.3111 | 0.3408 | 0.3202 | 0.3259 | 0.3069 | 0.3278 | 0.309 | 0.3963 | 0.4156 | 0.3935 | 0.4652 |
| | Average | 0.2984 | 0.2441 | 0.2623 | 0.2235 | 0.2779 | 0.2367 | 0.2639 | 0.2248 | 0.2616 | 0.2217 | 0.3206 | 0.3007 | 0.3193 | 0.3525 |
| Exchange | H96 | 0.2019 | 0.0814 | 0.2006 | 0.0839 | 0.2101 | 0.0886 | 0.1996 | 0.0825 | 0.2042 | 0.0855 | 0.1986 | 0.0823 | 0.1964 | 0.0811 |
| | H192 | 0.2838 | 0.1547 | 0.3132 | 0.193 | 0.3035 | 0.1805 | 0.2975 | 0.1714 | 0.3018 | 0.1801 | 0.2889 | 0.1678 | 0.2887 | 0.1671 |
| | H336 | 0.4237 | 0.3088 | 0.4129 | 0.3227 | 0.4267 | 0.3454 | 0.4366 | 0.3592 | 0.4254 | 0.3163 | 0.4058 | 0.3163 | 0.3978 | 0.3057 |
| | H720 | 0.5699 | 0.5139 | 0.6921 | 0.8465 | 0.6967 | 0.843 | 0.704 | 0.9066 | 0.7046 | 0.8659 | 0.6872 | 0.8138 | 0.6764 | 0.8101 |
| | Average | 0.3698 | 0.2647 | 0.4047 | 0.3615 | 0.4092 | 0.3644 | 0.4094 | 0.3799 | 0.409 | 0.3703 | 0.3952 | 0.3451 | 0.3898 | 0.341 |
| MotorImagery | H96 | 0.9106 | 2.7281 | 0.779 | 2.0277 | 0.159 | 0.438 | 0.7213 | 1.9562 | 0.3064 | 1.1502 | 1.1658 | 4.7017 | 0.8701 | 2.1921 |
| | H192 | 1.1498 | 4.5506 | 0.9345 | 3.289 | 0.2382 | 0.7093 | 0.9367 | 3.4629 | 0.8567 | 3.1102 | 1.3779 | 6.4838 | 1.04 | 3.4885 |
| | H336 | 1.3268 | 5.9559 | 1.0875 | 4.7769 | 0.3848 | 2.0705 | 1.1051 | 4.9205 | 1.1459 | 5.0606 | 1.4519 | 7.0403 | 1.2109 | 5.0798 |
| | H720 | 1.341 | 5.0979 | 1.2005 | 4.8849 | 0.5693 | 2.7657 | 1.2203 | 4.9106 | 1.2643 | 4.9838 | 1.2894 | 5.047 | 1.3409 | 5.305 |
| | Average | 1.1821 | 4.5831 | 1.0004 | 3.7446 | 0.3378 | 1.4959 | 0.9958 | 3.8126 | 0.8933 | 3.5762 | 1.3213 | 5.8182 | 1.1155 | 4.0164 |
| TDBrain | H96 | 0.6499 | 0.7692 | 0.5951 | 0.662 | 0.5966 | 0.6704 | 0.5916 | 0.6636 | 0.5906 | 0.6566 | 0.675 | 0.8627 | 0.764 | 1.106 |
| | H192 | 0.7463 | 1.0021 | 0.6689 | 0.8335 | 0.6653 | 0.8277 | 0.6657 | 0.8296 | 0.6626 | 0.8224 | 0.7387 | 1.0133 | 0.8257 | 1.2615 |
| | H336 | 0.846 | 1.2862 | 0.7467 | 1.0467 | 0.7451 | 1.0438 | 0.7469 | 1.0493 | 0.743 | 1.0373 | 0.809 | 1.2072 | 0.8951 | 1.4667 |
| | H720 | 0.9633 | 1.5443 | 0.8786 | 1.3606 | 0.8753 | 1.3551 | 0.8755 | 1.3545 | 0.871 | 1.3441 | 0.9204 | 1.495 | 1.0072 | 1.7817 |
| | Average | 0.8014 | 1.1505 | 0.7223 | 0.9757 | 0.7206 | 0.9742 | 0.7199 | 0.9742 | 0.7168 | 0.9651 | 0.7858 | 1.1445 | 0.873 | 1.404 |
| BeijingAir | H96 | 0.4399 | 0.5279 | 0.428 | 0.5181 | 0.4326 | 0.5145 | 0.4411 | 0.5275 | 0.43 | 0.5219 | 0.5041 | 0.6652 | 0.5983 | 1.1597 |
| | H192 | 0.4634 | 0.5687 | 0.4539 | 0.5716 | 0.4578 | 0.5677 | 0.4674 | 0.5763 | 0.4483 | 0.5627 | 0.5406 | 0.7403 | 0.6461 | 1.214 |
| | H336 | 0.4731 | 0.591 | 0.466 | 0.5905 | 0.4673 | 0.5933 | 0.4664 | 0.5884 | 0.4639 | 0.5862 | 0.5477 | 0.7559 | 0.6281 | 1.077 |
| | H720 | 0.5068 | 0.6415 | 0.4858 | 0.6231 | 0.487 | 0.6123 | 0.4931 | 0.6292 | 0.4944 | 0.6354 | 0.5706 | 0.7879 | 0.6531 | 1.1493 |
| | Average | 0.4708 | 0.5823 | 0.4584 | 0.5758 | 0.4612 | 0.5719 | 0.467 | 0.5804 | 0.4591 | 0.5765 | 0.5408 | 0.7373 | 0.6314 | 1.15 |
| BenzeneConcentration | H96 | 0.0144 | 0.0065 | 0.0393 | 0.0093 | 0.0482 | 0.0088 | 0.0132 | 0.0059 | 0.0387 | 0.0093 | 0.7636 | 0.9311 | 0.8622 | 1.2643 |
| | H192 | 0.0134 | 0.0065 | 0.0237 | 0.0095 | 0.0354 | 0.0076 | 0.0164 | 0.0062 | 0.0329 | 0.0101 | 0.81 | 1.0032 | 0.9001 | 1.3454 |
| | H336 | 0.0173 | 0.008 | 0.0396 | 0.0093 | 0.0551 | 0.0126 | 0.0196 | 0.0075 | 0.03 | 0.0113 | 0.8294 | 1.0425 | 0.8703 | 1.2813 |
| | H720 | 0.026 | 0.0114 | 0.0504 | 0.0147 | 0.0529 | 0.0143 | 0.0258 | 0.0125 | 0.0279 | 0.0141 | 0.8564 | 1.104 | 0.8705 | 1.281 |
| | Average | 0.0178 | 0.0081 | 0.0383 | 0.0107 | 0.0479 | 0.0108 | 0.0188 | 0.008 | 0.0324 | 0.0112 | 0.8149 | 1.0202 | 0.8758 | 1.293 |
| AustraliaRainfall | H96 | 0.7288 | 0.805 | 0.7309 | 0.8139 | 0.7265 | 0.8084 | 0.7276 | 0.8105 | 0.7263 | 0.8061 | 1.0919 | 1.9107 | 1.0833 | 1.8775 |
| | H192 | 0.7505 | 0.8387 | 0.7562 | 0.8554 | 0.7536 | 0.8489 | 0.7543 | 0.8516 | 0.7524 | 0.8469 | 1.1221 | 1.9776 | 1.1147 | 1.9491 |
| | H336 | 0.7592 | 0.8495 | 0.7635 | 0.8635 | 0.7622 | 0.8616 | 0.7639 | 0.8653 | 0.7614 | 0.8593 | 1.1276 | 1.9762 | 1.1191 | 1.9442 |
| | H720 | 0.7642 | 0.8573 | 0.7717 | 0.8769 | 0.7699 | 0.8739 | 0.7699 | 0.8737 | 0.7688 | 0.8707 | 1.1406 | 2.0112 | 1.1319 | 1.9792 |
| | Average | 0.7507 | 0.8376 | 0.7556 | 0.8524 | 0.7531 | 0.8482 | 0.7539 | 0.8503 | 0.7522 | 0.8457 | 1.1206 | 1.9689 | 1.1123 | 1.9375 |
| KDDCup2018 | H96 | 0.6265 | 1.0816 | 0.6472 | 1.1398 | 0.6537 | 1.1833 | 0.627 | 1.1126 | 0.645 | 1.1106 | 0.6611 | 1.1519 | 0.7816 | 1.5059 |
| | H192 | 0.6263 | 0.9778 | 0.6452 | 1.09 | 0.6462 | 1.0868 | 0.6361 | 1.0437 | 0.6281 | 1.034 | 0.6584 | 1.0689 | 0.7744 | 1.4141 |
| | H336 | 0.6409 | 1.006 | 0.6538 | 1.0847 | 0.6495 | 1.0587 | 0.6286 | 0.9986 | 0.6303 | 0.9986 | 0.654 | 1.0403 | 0.7824 | 2.0965 |
| | H720 | 0.6159 | 0.8894 | 0.6308 | 0.9885 | 0.6348 | 1.0067 | 0.6174 | 0.9563 | 0.6286 | 0.9754 | 0.6812 | 1.0882 | 0.8212 | 1.4917 |
| | Average | 0.6274 | 0.9887 | 0.6442 | 1.0757 | 0.6461 | 1.0839 | 0.6273 | 1.0302 | 0.633 | 1.0297 | 0.6637 | 1.0873 | 0.7899 | 1.6271 |
| PedestrianCounts | H96 | 0.2572 | 0.2383 | 0.2519 | 0.2182 | 0.2415 | 0.2199 | 0.2391 | 0.2188 | 0.2415 | 0.2157 | 1.0932 | 2.6555 | 0.9544 | 1.9948 |
| | H192 | 0.2715 | 0.2654 | 0.2731 | 0.2555 | 0.2545 | 0.2529 | 0.2656 | 0.2547 | 0.3308 | 0.3223 | 1.0999 | 2.6442 | 0.9645 | 2.029 |
| | H336 | 0.2945 | 0.3068 | 0.3008 | 0.3026 | 0.3023 | 0.3086 | 0.2827 | 0.2945 | 0.2825 | 0.2839 | 1.123 | 2.6634 | 0.9879 | 2.0655 |
| | H720 | 0.3318 | 0.3812 | 0.3388 | 0.3778 | 0.3324 | 0.383 | 0.328 | 0.3755 | 0.3245 | 0.3585 | 1.2082 | 3.0112 | 1.0517 | 2.2378 |
| | Average | 0.2888 | 0.2979 | 0.2912 | 0.2885 | 0.2827 | 0.2911 | 0.2789 | 0.2859 | 0.2948 | 0.2951 | 1.1311 | 2.7436 | 0.9896 | 2.0818 |
| Average | Average | 0.4784 | 0.7592 | 0.4549 | 0.6857 | 0.4104 | 0.5288 | 0.4524 | 0.6906 | 0.4478 | 0.673 | 0.7258 | 1.4186 | 0.746 | 1.3919 |
| Rank | Average | 3.29 | 3.43 | 3.21 | 2.79 | 3.43 | 3.21 | 2.79 | 3.07 | 3.07 | 3.29 | 5.93 | 5.93 | 6.29 | 6.29 |

Table 14: **Full results iPatch.** We present TSLib (left) and UTSD datasets (right). See Tab. 4 for average MSE, MAE, and Rank.

| Dataset | Model | iPatch | | | | Dataset | Model | iPatch | | | |
| --- | --- | --- | --- | --- | --- | --- | --- | --- | --- | --- | --- |
| | Metric | MAE | | MSE | | | Metric | MAE | | MSE | |
| | Statistic | Mean | Min | Mean | Min | | Statistic | Mean | Min | Mean | Min |
| ETTh1 | 96 | 0.4074 | 0.3995 | 0.3877 | 0.379 | MotorImagery | 96 | 0.1433 | 0.1287 | 0.4238 | 0.3541 |
| | 192 | 0.4241 | 0.4221 | 0.415 | 0.413 | | 192 | 0.2633 | 0.2079 | 1.0616 | 0.7352 |
| | 336 | 0.4301 | 0.4284 | 0.4241 | 0.4198 | | 336 | 0.3999 | 0.3458 | 2.183 | 1.8659 |
| | 720 | 0.4635 | 0.4599 | 0.45 | 0.4451 | | 720 | 1.1959 | 1.1879 | 4.8701 | 4.8506 |
| | Average | 0.4313 | 0.4275 | 0.4192 | 0.4142 | | Average | 0.5006 | 0.4676 | 2.1346 | 1.9515 |
| ETTm1 | 96 | 0.383 | 0.3726 | 0.3229 | 0.3115 | TDBrain | 96 | 0.5985 | 0.5909 | 0.6747 | 0.6592 |
| | 192 | 0.418 | 0.4171 | 0.3749 | 0.3719 | | 192 | 0.6704 | 0.665 | 0.8383 | 0.824 |
| | 336 | 0.4478 | 0.446 | 0.4289 | 0.4259 | | 336 | 0.7483 | 0.7423 | 1.0502 | 1.036 |
| | 720 | 0.4963 | 0.4919 | 0.4937 | 0.4862 | | 720 | 0.871 | 0.8704 | 1.3459 | 1.3451 |
| | Average | 0.4363 | 0.4319 | 0.4051 | 0.3989 | | Average | 0.722 | 0.7171 | 0.9773 | 0.9661 |
| ETTh2 | 96 | 0.3445 | 0.3423 | 0.2935 | 0.2902 | BeijingAir | 96 | 0.4345 | 0.4302 | 0.5334 | 0.5231 |
| | 192 | 0.3953 | 0.3931 | 0.3759 | 0.3711 | | 192 | 0.4559 | 0.455 | 0.569 | 0.5643 |
| | 336 | 0.4337 | 0.4323 | 0.4226 | 0.4205 | | 336 | 0.4665 | 0.4658 | 0.5962 | 0.5941 |
| | 720 | 0.4443 | 0.4419 | 0.4197 | 0.4172 | | 720 | 0.4721 | 0.4685 | 0.5909 | 0.5843 |
| | Average | 0.4045 | 0.4024 | 0.3779 | 0.3747 | | Average | 0.4573 | 0.4549 | 0.5724 | 0.5665 |
| ETTm2 | 96 | 0.2282 | 0.2281 | 0.1145 | 0.1134 | BenzeneConcentration | 96 | 0.0197 | 0.0192 | 0.0059 | 0.0059 |
| | 192 | 0.2595 | 0.2593 | 0.146 | 0.1457 | | 192 | 0.0247 | 0.0244 | 0.0112 | 0.0111 |
| | 336 | 0.2865 | 0.285 | 0.18 | 0.1786 | | 336 | 0.0188 | 0.0166 | 0.0077 | 0.0076 |
| | 720 | 0.3181 | 0.3148 | 0.214 | 0.2099 | | 720 | 0.0315 | 0.0302 | 0.0152 | 0.0149 |
| | Average | 0.2731 | 0.2718 | 0.1636 | 0.1619 | | Average | 0.0237 | 0.0226 | 0.01 | 0.0098 |
| Electricity | 96 | 0.2315 | 0.2304 | 0.1333 | 0.133 | AustraliaRainfall | 96 | 0.7284 | 0.7274 | 0.8115 | 0.8093 |
| | 192 | 0.2471 | 0.2456 | 0.1511 | 0.15 | | 192 | 0.7533 | 0.7524 | 0.85 | 0.8484 |
| | 336 | 0.2808 | 0.2782 | 0.1798 | 0.1769 | | 336 | 0.7623 | 0.7608 | 0.8624 | 0.8583 |
| | 720 | 0.3196 | 0.3159 | 0.2295 | 0.2268 | | 720 | 0.7691 | 0.7683 | 0.8719 | 0.8702 |
| | Average | 0.2698 | 0.2675 | 0.1734 | 0.1717 | | Average | 0.7533 | 0.7522 | 0.8489 | 0.8466 |
| Weather | 96 | 0.204 | 0.2031 | 0.1532 | 0.1525 | KDDCup2018 | 96 | 0.6556 | 0.6539 | 1.1853 | 1.1828 |
| | 192 | 0.2501 | 0.2494 | 0.2032 | 0.2016 | | 192 | 0.6455 | 0.6454 | 1.0951 | 1.0948 |
| | 336 | 0.2879 | 0.2878 | 0.2516 | 0.2511 | | 336 | 0.6397 | 0.6366 | 1.0313 | 1.0267 |
| | 720 | 0.3348 | 0.3337 | 0.3202 | 0.3192 | | 720 | 0.6504 | 0.6478 | 1.0057 | 0.9874 |
| | Average | 0.2692 | 0.2685 | 0.232 | 0.2311 | | Average | 0.6478 | 0.6459 | 1.0793 | 1.0729 |
| Exchange | 96 | 0.2146 | 0.2114 | 0.0934 | 0.0906 | PedestrianCounts | 96 | 0.2316 | 0.2309 | 0.2139 | 0.213 |
| | 192 | 0.3127 | 0.3067 | 0.1901 | 0.1853 | | 192 | 0.2628 | 0.2498 | 0.2564 | 0.2475 |
| | 336 | 0.4260 | 0.4219 | 0.3461 | 0.3397 | | 336 | 0.2797 | 0.274 | 0.2939 | 0.2894 |
| | 720 | 0.8205 | 0.8104 | 1.1578 | 1.1318 | | 720 | 0.3196 | 0.318 | 0.3726 | 0.372 |
| | Average | 0.4435 | 0.4376 | 0.4469 | 0.4369 | | Average | 0.2734 | 0.2682 | 0.2842 | 0.2805 |

