# OpenReview forum: "There are no Champions in Long-Term Time Series Forecasting"
_NeurIPS.cc/2025/Position_Paper_Track — Submitted to NeurIPS 2025 Position Paper Track_

### Official Review · Reviewer_ynSr · 2025-08-07

**Significance:** 2
**Presentation:** 4
**Rating:** 7
**Confidence:** 3

**Summary:**

The paper shows the important fact that any time series forecasting method on current time series benchmarks, as long as we artifitially choose some 'good' metric and story-telling method, can be tagged as 'champion' compared to baselines; also, current 'model selection' methods are unreliable. Thus, authors argue future benchmarks to improve benchmarks through different approaches; future papers to avoid  wrong claims; and our community to revise model-selection guidelines.

**Strengths:**

This paper is well-proven with thorough experiments. For example, the authors show possible 'champion cooking-up story-telling method' in these aspects,
(1) plot drawing
(2) Friedman-test-same model achieve questionable champion on MAE
(3) Limited gains of model complexity
(4) Hyper Parmeter sensitivity
(5) Metric sensitivity.

For model selection methods, authors show that:
(1) Model rankings are highly sensitive to dataset and horizon selection through 5000 repeated experiements.
(2) Current model selection methods need improvement.

Overall, this paper has conducted very thorough and solid experiments from various aspects, supporting its claims quite well.

**Weaknesses:**

The authors have discussed very thoroughly how fragile current benchmarks are with respect to hyperparameters and evaluation pipelins, which is good. However, current benchmarks are lacking in quite many domains. For example, (1) traditional methods may outperform current 'sota' DL-based methods, implying DL methods are only learning trival things; (2) lack of related contexts (e.g. perhaps news for finance, perhaps geometry information for weather forecasting).

**Questions:**

The authors have proposed the lack of current TSF benchmarks. I am perhaps more aggressive than the authors. I am in the TSF community, but I wonder: whether general-purpose TSF makes sense?

Let me give an example. The best weather forecasting AI method is probably using some 3D-ViT based methods with multimodality inputs. The best financial AI method is perhaps some ensemble of Features+Trees, some end2end methods. If there does not exist such 'general' TSF model surpassing these domain-specific sotas, is so-called 'general' TSF model worth studying? If we view weather as time series, if we view finance as time series, if we view NLP as time series, does it mean that we can find some TSF-based methods that can surpass GPT/Gemini/etc. on NLP, surpass best QT firms on Finance, and surpass the best methods for weather forecasting?

**Alternative Position:**

Yes, and alternative positions are well-considered and named but not addressed

**Author Identification:**

No.

**Context:**

3

**Discussion:**

3

**Ethics:**

["NO or VERY MINOR ethics concerns only"]

**Position:**

Yes, the paper argues for or against a position related to machine learning.

**Support:**

4

**Thoroughness:**

4

---

### Official Review · Reviewer_GjiJ · 2025-08-09

**Significance:** 3
**Presentation:** 3
**Rating:** 4
**Confidence:** 4

**Summary:**

This paper advocates the position that there is no absolute best model in time series forecasting domain at the moment and the performance of the model varies across dataset and often differs based on what evaluation metric to use to measure performance. The authors recommends that, instead of developing more and more complex time series forecasting model, the researchers should focus on improving benchmarking standards with more rigorous evaluation metrics, statistical testing or comprehensive experimental design.

**Strengths:**

1. The paper present a clear, concise and well articulated position. The position is timely and would spark discussions in the domain of time series forecasting.
2. The authors emphasis on the need of better benchmarking through more diverse and relevant real-world time series forecasting datasets, just like ImageNet in Image Classification domain, is convincing.
3. The focus on more statistical testing to find out best performing model is an important recommendation for time series forecasting domain.
4. The authors experimentally proven that more complex model does not necessarily outperform simpler models in many challenging use cases.
5. The authors provide a detailed experimental analysis to show how sensitive the state of the art models are across the dataset and why the future studies need more focus on robust hyper-parameter tuning and re-producible experiments in the time series forecasting domain.

**Weaknesses:**

1. This paper started the initial discussion with a strong and bold position. However, the arguments and recommendations made by the authors did not comprehensively and completely develop throughout the paper.
2. Even though the authors highlight the loopholes or issues with existing evaluation metrics widely used in the time series forecasting community, they did not strongly recommend why this metric fails technically and what should be the exact technical way of using such metrics in future time series forecasting research.
3. The authors spot on the need for more representative time series forecasting datasets, but they don't really guide the researchers on how to build those more robust datasets, what to avoid when developing those datasets, what to include, etc.
4. The authors tell us in the beginning that we should divert our focus on building more complex models because even simpler architectures like DLinear do pretty well across benchmarks, but it is not clear why more complex architectures are not able to do better (what is wrong fundamentally) and what we should keep in mind when designing next models.
Overall, the paper lacks depth in discussion and analysis.

**Questions:**

1. The foundation models research in the "Alternative Views" section was interesting to me at the start, but why did it end in a surface-level analysis without any detailed explanation and sufficient relevant evidence from the literature?
2. It is not clear to me if the authors at the end recommends the researchers to come up with more generalized time series foundation models which can outperforms with a significant margin. Or do the authors believe that it is simply not possible to design such methods in the near future because of the inherent diversity of the datasets coming from diverse domains?
3. After reading this paper, it is clear that the time series forecasting models are more or less indistinguishable in each and every dataset. But if that is true, then what should the future researchers do?
4. I did not find multiple strong and convincing pieces of evidence in the literature to employ the mentioned statistical tests to determine model superiority. I believe we need a more comprehensive and technical explanation to determine why the existing popular metrics are failing to identify the best model in a more robust manner. What's your take on this?

**Alternative Position:**

Yes, and alternative positions are well-considered and named but not addressed

**Author Identification:**

No.

**Context:**

3

**Discussion:**

3

**Ethics:**

["NO or VERY MINOR ethics concerns only"]

**Position:**

Yes, the paper argues for or against a position related to machine learning.

**Support:**

3

**Thoroughness:**

4

---

### Official Review · Reviewer_8GGG · 2025-08-18

**Significance:** 3
**Presentation:** 3
**Rating:** 10
**Confidence:** 3

**Summary:**

This paper offers a constructive evaluation of Long-Term Series Forecasting by conducting evaluations across 5 models from 14 different data sets. They found that although there are claims of continuous improvement, these claims rely on specific circumstances, and that no one model consistently outperforms the others. They offer recommendations of standardizing evaluations, increasing benchmarking robustness, and examining the impact of underlying data quality on model performance.

**Strengths:**

The strengths of this paper include strong arguments for its position and a large amount of testing to provide support for its position. The paper is timely and organized clearly.

**Weaknesses:**

NA this paper was comprehensive and rigorous.

**Questions:**

This paper covered many subject areas for application. Are there certain areas that would benefit more or are more immediately in need of more rigorous benchmarking and evaluations from the research?

**Alternative Position:**

Yes, and alternative positions are well-considered and named but not addressed

**Author Identification:**

No.

**Context:**

4

**Discussion:**

3

**Ethics:**

["NO or VERY MINOR ethics concerns only"]

**Position:**

Yes, the paper argues for or against a position related to machine learning.

**Support:**

3

**Thoroughness:**

4

---

### Note · Authors · 2025-09-01

**1-10 Additional Comments:**

We want to use this opportunity to stress again that a harmonization with the main track would improve the experience (deadlines, rebuttals, etc.).

**1-11 Submit Again:**

Probably yes

**1-1 Submission Process:**

3

**1-2 Next Year:**

To improve the process, we recommend minimizing delays where possible, providing a clear timeline from the outset, and, most importantly, incorporating a short rebuttal period.

**1-3 Future Development:**

The track would benefit from a small rebuttal following the NeurIPS main/benchmark tracks and the ICML position paper track. Indeed, while we understand that most of the discussion stimulated by the paper should happen at the conference, a rebuttal is necessary to signal possible misconceptions, misunderstandings, and mistakes directly to the involved reviewers. The current survey does not allow it.

**1-4 Interest:**

["Workshops for developing position papers"]

**1-5 Thoughtful:**

7

**1-6 Supportive:**

8

**1-7 Technical Aspects Versus Position:**

4

**1-8 Gate Keeping:**

9

**1-9 Camera Ready Changes:**

For the camera-ready version, we will perform minor revisions to clarify our position on the domain-specialist versus generalist model debate raised by reviewers GjiJ and ynSr. We agree with the reviewers that domain-specialized models will remain more effective and efficient than general forecasting models for practical applications. However, we also highlight that research into general forecasting models is valuable for uncovering methodological principles and informing the development of such specialized approaches. In this context, we will also revise the Alternative Views section to more clearly reflect the role of multi-modal models that can incorporate additional context from various domains.

For the remaining weaknesses and points raised, in particular from Reviewer GjiJ, we provide in our reviewer responses a detailed explanation of why they fall outside the scope of our paper.

**3-1 Review Response1:**

8GGG

**3-2 Reaction To Review1:**

We are honoured by the excellent score and thank the reviewer. We appreciate the reviewer’s strong agreement with our position, as well as the recognition that the paper’s strengths include well-supported arguments and extensive testing. We are also pleased that the paper was found to be timely and clearly organized.

The reviewer asks whether certain domains are more in need of rigorous benchmarking. We believe all domains would benefit, since improved benchmarks can help assess raw forecasting performance and guide model selection across datasets with varying noise, stationarity, and forecastability. For the sake of enlarging the stimulating discussion brought by the reviewer's question, we believe that to rigorously evaluate real-world performance in specific applications, domain-specific evaluation is ultimately required. For instance, current benchmarks mainly test how well models can predict N fixed timesteps ahead, without considering whether such forecasts are meaningful — whether they are clinically actionable in health, lead to gains in finance, or support reliable planning in energy or environmental management.

**3-3 Review Response2:**

GjiJ

**3-4 Reaction To Review2:**

We appreciate the reviewer’s agreement with our position and for recognizing that it is presented clearly, concisely, timely, and in a well-articulated manner. From a technical aspect, the reviewer appreciates the large amount of testing we provided. However, we believe that the technical expectations of the reviewer go beyond the scope of the position paper, which could have led to a low score that translates to “borderline reject” regardless of the support for our position.

More in detail, the raised weaknesses on metrics, datasets, and model design request “how-to” recommendations better suited for technical papers. Our goal was to highlight current limitations in LTSF benchmarking, provide empirical evidence (acknowledged by all reviewers), and outline community-level recommendations to improve transparency, rigor, and practical relevance (Sec. 5). While we agree that questions such as “why complex models underperform” are important, they require dedicated studies, and we believe our position paper will inspire such future work in the style of the recently published [A].

Finally, regarding the asked questions (Q):

Q1. We believe our Alternative Views section is quite broad in scope and aims to acknowledge rather than exhaustively analyze each perspective, considering also the limited space suggested by the conference guidelines (a single section).
Q2. Please refer to the answer to Reviewer ynSr.
Q3. We dedicated Section 5 to recommendations for future LTSF research.
Q4. The mentioned statistical tests are recommended by (Demšar, 2006) (cited over 16k times). Statistical tests substantiate claims regarding model superiority.

[A] Chen et al., A Closer Look at Transformers for Time Series Forecasting: Understanding Why They Work and Where They Struggle, ICML 2025

**3-5 Review Response3:**

ynSr

**3-6 Reaction To Review3:**

We appreciate the reviewer’s agreement with our position and the note that we have conducted very thorough and solid experiments from various aspects to support our position.

We acknowledge the reviewer’s observation that current benchmarks may fail to capture the strengths of traditional methods in certain domains. We believe this point is partly addressed in our Section 4.2 and Appendix H. In Section 4.2, we show that excluding baselines (e.g., potentially a traditional method) may expectably crown other models. While in Appendix H, we extended the evaluation to traditional methods such as ARIMA and CLOF, showing that on Exchange (stock market predictions), they outperform all other baselines.

Responding to the reviewer’s question, we acknowledge that a universal “general-purpose” TSF model may have limited practical importance, as domains like weather or finance benefit from models that exploit rich contextual information. Our position is that such models are valuable primarily in the context of basic research, where they can drive methodological advances that later support domain-specific innovations. At the same time, we emphasize that current LTSF evaluation practices often create a misleading impression of universal champions. We therefore advocate for more rigorous and transparent evaluations, while recognizing that domain-specialized models will likely remain the most effective in practice. To make this clearer, we will revise the Alternative Views section to explicitly address the role of domain-specific models, as also touched upon in our discussion of multimodal LLMs.

---

### Meta-Review · Area_Chair_gW43 · 2025-09-15

**Rating:** 4
**Confidence:** 4

**Strengths:**

+ The paper aims to discuss an important problem in time series forecasting
+ Rigorous statistics testing and thorough analysis have been performed to analyze the experiments

**Weaknesses:**

- Only a subset of the state-of-art models have been tested on a very long but "toy" time series datasets (i.e., they might not be a good representative of the data in practical applications)

- While the analysis is interesting, the results are not surprising: long-term time series forecasting is known to be extremely difficult, with large-variance in the prediction.

- The paper makes bold claims and position at the beginning of the paper, but they are not well-supported in the paper. There is also no insights into how to better address the long-term forecasting.

**Questions:**

There has been a lot of development in time series foundation models for forecasting. Have you tested these models?

**Ethics:**

There are no ethical violations or concerns raised by reviewers about this paper.

**Thoroughness:**

3

---

### Decision · Program_Chairs · 2025-09-26

Reject